# Toward a Complete Criterion for Value of Information in Insoluble Decision Problems

**Ryan Carey**                                                                                   *ry.duff@gmail.com*
*Oxford University*

**Sanghack Lee**                                                                           *sanghack@snu.ac.kr*
*Seoul National University*

**Robin Evans**                                                                           *evans@stats.ox.ac.uk*
*Oxford University*

**Reviewed on OpenReview:** *https://openreview.net/forum?id=0RUzRVO5Jn*

## Abstract

In a decision problem, observations are said to be material if they must be taken into account to perform optimally. Decision problems have an underlying (graphical) causal structure, which may sometimes be used to evaluate certain observations as immaterial. For soluble graphs — ones where important past observations are remembered — there is a complete graphical criterion; one that rules out materiality whenever this can be done on the basis of the graphical structure alone. In this work, we analyse a proposed criterion for insoluble graphs. In particular, we prove that some of the conditions used to prove immateriality are necessary; when they are not satisfied, materiality is possible. We discuss possible avenues and obstacles to proving necessity of the remaining conditions.

## 1 Introduction

We can view any decision problem as having an underlying causal structure — a graph consisting of chance events, decisions and utility variables (or "outcomes"), and their causal relationships. Sometimes, it is possible to evaluate key aspects of a decision problem from its causal structure alone. For example, in Figure 1a and Figure 1b, we see two causal structures. In each graph, there is an observation $Z$, which is a parent of the decision $X$, which affects the outcome $Y$. The difference is that in Figure 1b, $Z$ also directly influences $Y$, whereas in Figure 1a, it does not.

To fully describe these decision problems we must specify, for each non-decision variable, a probability distribution. Each probability distribution must be conditional only on the outcomes of the directed causes (i.e. the parent variables), a condition known as Markov compatibility. In Figure 1b, a Markov compatible decision problem is shown, where the variable $Z$ is a Bernoulli trial (i.e. a coin flip), and the decision-maker is rewarded with $Y = 1$ if they state the outcome of $Z$ (i.e. call the outcome of the coin flip), otherwise the reward is $Y = 0$. In this scenario, a greater reward can be attained if the decision is allowed to be conditioned on $Z$, and so $Z$ is said to be material. Specifically, if it is possible to observe the Bernoulli trial before selecting $X$, then one can attain a utility of 1 surely, whereas if one cannot observe the Bernoulli trial, the maximum expected utility is 0.5.

For the causal structure shown in Figure 1a, however, there exists an optimal decision rule that ignores the value of $Z = z$ entirely, in other words $Z$ is immaterial; this holds true for any decision problem compatible with the graph. This is evident because given any decision $X = x$, the observation $Z$ is independent of $Y$, and so there is no need for the decision to vary with $Z$. (This can be proved from the fact that $Z$ is d-separated from $Y$ given $X$.)

There are several reasons that we may want to evaluate whether a graph allows $Z$ to be material. Firstly, for algorithmic efficiency — if an observed variable is immaterial, then the optimal policies are contained in a small subset of all available policies, that we can search exponentially more quickly. (For example, in Figure 1a, there are four deterministic mappings from $Z$ to $X$, but if we let $X$ ignore $Z$, then there are just two possible choices, $X = 1$ and $X = 0$.)

Secondly, materiality can have consequences for the fairness of a decision-making procedure. Suppose that $Z$ designates the gender of candidates available to a recruiter, which are male ($Z = 1$) or female ($Z = 0$) with equal probability, while $X$ indicates whether that person is ($X = 1$) or is not ($X = 0$) recruited, and $Y$ indicates whether that person is ($Y = 1$) or is not ($Y = 0$) hired. If $Y$ is correlated with $Z$ given $X$, then the applicant's gender is material for the recruiter, and to maximize the hiring probability, they will have to recruit applicants at different rates based on their gender. If the causal structure is that of Figure 1a, then materiality can be ruled out, meaning that unfair behaviour is not necessary for optimal performance, whereas the causal structure of Figure 1b can incentivize unfairness. Such an analysis can also be extended to counterfactual fairness (Kusner et al., 2017): in an arbitrary graph where $Z$ is a sensitive variable (such as gender), counterfactual fairness can arise only when there is a path $Z \rightarrow \cdots \rightarrow O \rightarrow X$, where the observation $O$ is material (Everitt et al., 2021).

Thirdly, materiality can have implications for AI safety — if $Z$ represents a corrective instruction from a human overseer, and there exists no path $Z \rightarrow \cdots \rightarrow O \rightarrow X$ where $O$ is material, then there exist optimal policies that ignore this instruction (Everitt et al., 2021). Materiality is also relevant for evaluations of agents' intent (Halpern & Kleiman-Weiner, 2018; Ward et al., 2024b), and relatedly, their incentives to control parts of the environment (Everitt et al., 2021; Farquhar et al., 2022). For an agent to intentionally manipulate a variable $Z$ to obtain an outcome $Y = y$, there must be a path $p : X \rightarrow \cdots \rightarrow Z \rightarrow \cdots \rightarrow Y$ where for each of its decisions $X'$ lying on $p$, the parent $O'$ along $p$ is material for $X'$. In general, a stronger criterion for ruling out materiality will allow us to rule out unfair or unsafe behaviour for a wider range of agent-environment interactions (Everitt et al., 2021).

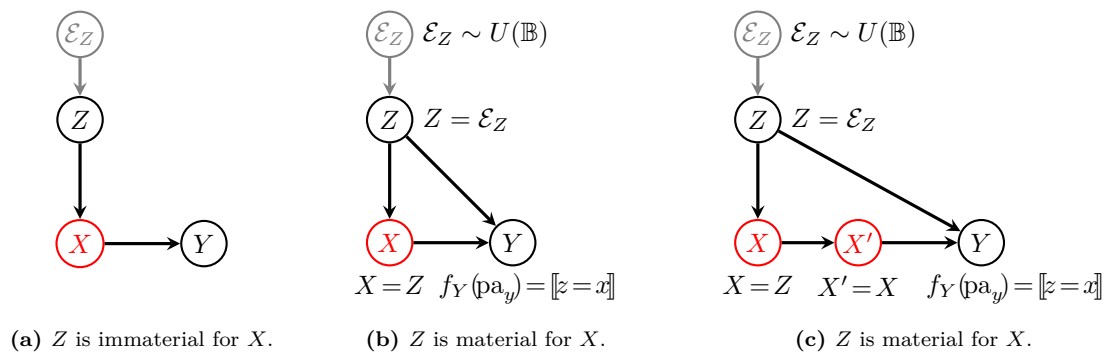

**(a)** $Z$ is immaterial for $X$.      **(b)** $Z$ is material for $X$.      **(c)** $Z$ is material for $X$.

**Figure 1:** Three graphs, with decisions in red, and a real-valued outcome $Y$. We write $U(\mathbb{B})$ for a uniform distribution over $\mathbb{B}$, i.e. a Bernoulli distribution with $p = 0.5$.

Any procedure for establishing immateriality based on the causal structure may be called a *graphical criterion*. For example, if a decision $X$ is not an ancestor of the outcome $Y$, then all of the variables observed at $X$ are immaterial. An ideal graphical criterion would be proved *complete*, in that it can establish immateriality whenever this is possible from the graphical structure alone. Clearly, this criterion is not complete, because in Figure 1a, $X$ is an ancestor of the outcome, but we still proved $Z$ immaterial. So far, a graphical criterion from van Merwijk et al. (2022) has been proved complete, but only under some significant restrictions. The causal structure must be *soluble*, meaning that all of the important information observed from past decisions is remembered at later decision points. Also, no criteria has been proved complete for identifying immaterial decisions, i.e. past decisions that can be safely forgotten.

For insoluble graphs, there is the criterion of Lee & Bareinboim (2020, Thm. 2), which can identify immaterial decisions and is (strictly) more potent in general. However, it is not yet known whether this criterion is complete. In particular, it is not yet clear whether several of its conditions are necessary. For example, one

case where all existing criteria are silent is the simple graph shown in Figure 1c — we would like to know whether we can rule out $X$ being a material observation for $X'$. We cannot use van Merwijk et al. (2022) because $X$ is a decision, and because the graph is insoluble.[1] Furthermore, we cannot establish immateriality using Lee & Bareinboim (2020, Thm. 2), because it violates a property that we term LB-factorizability, which we will discuss in Section 3.4.[2]

By studying Figure 1c in a bespoke fashion, we find that there exists a decision problem with the given causal structure, where $X$ is material for $X'$. As shown in Figure 1c, $Z$ is a Bernoulli variable, and $Y$ is equal to 1 if $Z = X'$ and to 0 otherwise. If $X$ is observed by $X'$, then a reward of $\mathbb{E}[Y] = 1$ can be achieved by the policy $X' = X = Z$. If $X$ is not observed, the greatest achievable reward is lower, at $\mathbb{E}[Y] = 0.5$, implying materiality.

This raises a question: by generalizing this construction, can we prove that requirement I of LB-factorizability is necessary to prove immateriality for a wide class of graphs? This work will prove that this requirement is indeed necessary, meaning that materiality cannot be excluded for a wide class of graphs including Figure 1c.

It remains an open question whether the criterion of Lee & Bareinboim (2020, Thm. 2) as a whole is complete, in that its other conditions are necessary for establishing immateriality. In the case that it is complete, our work is a step toward proving this. On the other hand, we also present some graphs where materiality is difficult to establish, that — if the criterion is not complete — could bring us closer to a proof of incompleteness.

The structure of the paper is as follows. In Section 2, we will recap the formalism used by Lee & Bareinboim (2020) for modelling decision problems, based on structural causal models. In Section 3, we will review existing procedures for proving that an observation can or cannot be material. In Section 4, we will establish our main result: that requirement I of LB-factorizability is necessary to establish immateriality. In Section 5, we present some analogous results for other requirements of LB-factorizability that could serve as a building block for proving the necessity of those requirements. We then illustrate the problems that arise in trying to prove necessity of those further requirements, and outline some possible directions for further work. Finally, in Section 6, we conclude.

## 2 Setup

Throughout the paper, we will define decision problems using structural causal models (Pearl, 2009, Chapter 7), following the formalism of Lee & Bareinboim (2020), although our results also apply equally to Bayesian networks and influence diagrams.

### 2.1 Structural causal models

A structural causal model (SCM) $\mathcal{M}$ is a tuple $\langle \boldsymbol{U}, \boldsymbol{V}, P(\boldsymbol{U}), \mathbf{F} \rangle$, where $\boldsymbol{U}$ is a set of variables determined by factors outside the model, called *exogenous* following a joint distribution $P(\boldsymbol{U})$, and $\boldsymbol{V}$ is a set of endogenous variables whose values are determined by a collection of functions $\mathbf{F} = \{f_V\}_{V \in \boldsymbol{V}}$ such that $V \leftarrow f_V(\mathrm{Pa}(V), \boldsymbol{U}_V)$ where $\mathrm{Pa}(V) \subseteq \boldsymbol{V} \setminus \{V\}$ is a set of endogenous variables and $\boldsymbol{U}_V \subseteq \boldsymbol{U}$ is a set of exogenous variables. The observational distribution $P(\boldsymbol{v})$ is defined as $\sum_{\boldsymbol{u}} \prod_{V \in \boldsymbol{V}} P(v|\mathbf{pa}_V, \boldsymbol{u}_V) P(\boldsymbol{u})$, where $\boldsymbol{u}_V$ is the assignment $\boldsymbol{u}$ restricted to variables $\boldsymbol{U}_V$. Furthermore, $\mathrm{do}(\boldsymbol{X} = \boldsymbol{x})$ represents the operation of fixing a set $\boldsymbol{X}$ to a constant $\boldsymbol{x}$ regardless of their original mechanisms. Such intervention induces a submodel $\mathcal{M}_{\boldsymbol{x}}$, which is $\mathcal{M}$ with $f_X$ replaced by $x$ for $X \in \boldsymbol{X}$. Then, an interventional distribution $P(\boldsymbol{v}\backslash\boldsymbol{x}|\mathrm{do}(\boldsymbol{x}))$ can be computed as the observational distribution in $\mathcal{M}_{\boldsymbol{x}}$. The induced graph of an SCM $\mathcal{M}$ is a DAG $\mathcal{G}$ on only the endogenous variables $\boldsymbol{V}$, where (i) $X \rightarrow Y$ if $X$ is an argument of $f_Y$; and (ii) $X \leftrightarrow Y$ if $\boldsymbol{U}_X$ and $\boldsymbol{U}_Y$ may be dependent, i.e. for any $\boldsymbol{u}_X, \boldsymbol{u}_Y, P(\boldsymbol{u}_X, \boldsymbol{u}_Y) \neq P(\boldsymbol{u}_X) \times P(\boldsymbol{u}_Y)$.

---

[1]Formally, this is because $W \not\perp Y \mid X \cup X'$, and $X' \not\perp Y \mid X \cup W$, as per the definition of solubility that we will review in Section 3.

[2]Specifically, requirement I of LB-factorizability is violated because $Y$ is d-connected to $\boldsymbol{\pi}_{X'}$ given $X'$.

We use the notation $\mathrm{Pa}(X)$, $\mathrm{Ch}(X)$, $\mathrm{Anc}(X)$ and $\mathrm{Desc}(X)$ to represent the parents, children, ancestors and descendants of a variable $X$, respectively, and take ancestors and descendants to include the node $X$ itself.[3]

We write $V_1{-}V_2$ to designate an edge whose direction may be $V_1 \to V_2$ or $V_1 \leftarrow V_2$. For a path $V_1 - \cdots - V_\ell$, we will use the shorthand $V_1 \text{ --- } V_\ell$, and for a directed path $V_1 \to \cdots \to V_\ell$, the shorthand $V_1 \dashrightarrow V_\ell$. For a path $p : A \text{ --- } B \text{ --- } C \text{ --- } D$, we will describe the segment $B \text{ --- } C$ using the shorthand $B \text{ -}^p\text{- } C$. We will use the shorthand $\boldsymbol{V}_{1:N}$ for a sequence of variables $V_1, \dots V_N$ indexed by $1, \dots, N$, $\boldsymbol{v}_{1:N}$ for a sequence of assignments, and $\boldsymbol{p}_{1:N}$ for a set of paths $p_1, \dots p_N$.

Some notations are used repeatedly when constructing causal models, such as tuples, bitstrings, indexing, and Iverson brackets. We will write a tuple as $z := \langle x, y \rangle$, and this may be indexed as $z[0] = x$. A bitstring of length $n$, i.e. a tuple of $n$ Booleans, may be written as $\mathbb{B}^n$, and a uniform distribution over this space, as $U(\mathbb{B}^n)$. We will denote a bitwise XOR operation by $\oplus$ so that, for example, $01 \oplus 11 = 10$. Bitstrings may also be used for indexing, for example, the $y^{\text{th}}$ bit of $x$ may be written as as $x[y]$, and the leftmost bits are of higher-order so that, for example, $0100[01] = 1$. Similarly, for random variables $X, Y$, we will write $X[Y]$ for a variable equal to $x[y]$ when $X = x$ and $Y = y$. Finally, the Iverson bracket $[\![P]\!]$ is equal to 1 if $P$ is true, and 0 otherwise.

## 2.2 Modelling decision problems

To turn an SCM into a decision problem, three further elements must be specified: a set of decision variables, a set of policies that the agent may use to control those decision variables, and a goal that the agent is trying to achieve.

A Mixed Policy Scope (Lee & Bareinboim, 2020) supplies the first two elements: identifying certain variables as decisions, and enumerating the *context variables* or "observations" $\boldsymbol{C}_X$ on which each decision $X$ is allowed to depend.

**Definition 1** (Mixed Policy Scope (MPS)). Given a DAG $\mathcal{G}$ on vertices $\boldsymbol{V}$, a *mixed policy scope* $\mathcal{S} = \langle X, \boldsymbol{C}_X \rangle_{X \in \boldsymbol{X}(\mathcal{S})}$ consists of a set of decisions $\boldsymbol{X}(\mathcal{S}) \subseteq \boldsymbol{V}$ and a set of context variables $\boldsymbol{C}_X \subseteq \boldsymbol{V}$ for each decision.

For a set of decisions $\boldsymbol{X}'$, we define their contexts as $\boldsymbol{C}_{\boldsymbol{X}'} = \bigcup_{X \in \boldsymbol{X}'} \boldsymbol{C}_X$.

A policy consists of a probability distribution for each decision $X$, conditional on its contexts $\boldsymbol{C}_X$.

**Definition 2** (Mixed Policy). Given an SCM $\mathcal{M}$ and scope $\mathcal{S} = \langle X, \boldsymbol{C}_X \rangle$, a *mixed policy* $\boldsymbol{\pi}$ (or a *policy*, for short) contains for each $X$ a decision rule $\pi_{X|\boldsymbol{C}_X}$, where $\pi_{X|\boldsymbol{C}_X} : \mathfrak{X}_X \times \mathfrak{X}_{\boldsymbol{C}_X} \mapsto [0,1]$ is a proper probability mapping from the domain $\mathfrak{X}_X$ of $X$ to the domain $\mathfrak{X}_{\boldsymbol{C}_X}$ of $\boldsymbol{C}_X$.[4]

We will say that such a policy $\boldsymbol{\pi}$ *follows* the scope $\mathcal{S}$, written $\boldsymbol{\pi} \sim \mathcal{S}$. A mixed policy is said to be *deterministic* if every decision is a deterministic function of its contexts.

Given a mixed policy scope, we obtain a new causal structure, described by a *scoped graph*.

**Definition 3** (Scoped graph). The *scoped graph* $\mathcal{G}_{\mathcal{S}}$ is obtained from $\mathcal{G}$ by replacing, for each decision $X \in \boldsymbol{X}(\mathcal{S})$, all inbound edges to $X$ with edges $C \to X$ for every $C \in \boldsymbol{C}_X$. We only consider scopes for which $\mathcal{G}_{\mathcal{S}}$ is acyclic.

In a scoped graph, we will always illustrate the decision variables as red circles, as shown in Figure 1 (Lee & Bareinboim, 2020). The final element of a decision problem — the agent's goal — will be represented by a real-valued variable $Y$. Throughout this paper, we will assume that $Y$ is neither a decision, nor a context.

The expected utility $\mu_{\boldsymbol{\pi}, \mathcal{S}}$ given a policy $\boldsymbol{\pi}$ is simply the expected value of $Y$ in the model $M_{\boldsymbol{\pi}}$, where each $f_X$ is replaced with $\pi_X$, i.e. $\mu_{\boldsymbol{\pi}, \mathcal{S}} := \mathbb{E}^{M_{\boldsymbol{\pi}}}[Y]$. When the scope is obvious, we will simply write $\mu_{\boldsymbol{\pi}}$.

---

[3]Note that $\mathrm{Pa}(X)$ is an intentional reuse of the notation used to describe the arguments of $f_X$ in the SCM definition, because the endogenous arguments of $f_X$ and the parents of $X$ in the induced graph are the same variables.

[4]Following Lee & Bareinboim (2020), we term this a "mixed policy" due to its including mixed strategies. Note that game theory also has a distinction between "mixed" policies, where the decision rules share a source of randomness, and "behavioural" policies, where they do not, and in this sense, the "mixed" policies of Lee & Bareinboim (2020) are actually *behavioural*.

This paper is concerned with materiality — whether removing one context variable from one decision will decrease the expected utility attainable by the best policy. We define materiality in terms of the value of information (Howard, 1990; Everitt et al., 2021).

**Definition 4** (Value of Information)**.** Given an SCM $\mathcal{M}$ and scope $\mathcal{S}$, the *maximum expected utility* (MEU) is $\mu^*_{\mathcal{S}} = \max_{\boldsymbol{\pi} \sim \mathcal{S}} \mu_{\boldsymbol{\pi}, \mathcal{S}}$. The *value of information* (VoI) of context $Z \in \boldsymbol{C}_X$ for decision $X \in \boldsymbol{X}(\mathcal{S})$ is $\mu^*_{\mathcal{S}} - \mu^*_{\mathcal{S}_{Z \not\to X}}$, where $\mathcal{S}_{Z \not\to X}$ is defined as $\langle X', \boldsymbol{C}_{X'} \rangle_{X' \in \boldsymbol{X}(\mathcal{S}) \setminus \{X\}} \cup \langle X, \boldsymbol{C}_X \setminus \{Z\} \rangle$.

The context $Z$ is *material* for $X$ in an SCM $\mathcal{M}$ if $Z$ has strictly positive value of information for $X$, otherwise it is *immaterial*.

### 2.3 Graphical criteria for independence

Knowing when variables are independent is an important step in identifying immaterial contexts, as we will discuss in the next section. Thus, we will make repeated use of d-separation, a graphical criterion that establishes the independence of variables in a graph.

**Definition 5** (d-separation; Verma & Pearl, 1988)**.** A path $p$ is said to be d-separated by a set of nodes $\boldsymbol{Z}$ if and only if:

1. $p$ contains a collider $X \to W \leftarrow Y$ such that the middle node $W$ is not in $\boldsymbol{Z}$ and no descendants of $W$ are in $\boldsymbol{Z}$, or
2. $p$ contains a chain $X \to W \to Y$ or fork $X \leftarrow W \to Y$ where $W$ is in $\boldsymbol{Z}$, or
3. one or both of the endpoints of $p$ is in $\boldsymbol{Z}$.

A set $\boldsymbol{Z}$ is said to d-separate $\boldsymbol{X}$ from $\boldsymbol{Y}$, written $(\boldsymbol{X} \perp_{\mathcal{G}} \boldsymbol{Y} \mid \boldsymbol{Z})$, if and only if $\boldsymbol{Z}$ d-separates every path from a node in $\boldsymbol{X}$ to a node in $\boldsymbol{Y}$. Sets that are not d-separated are called d-connected, written $\boldsymbol{X} \not\perp_{\mathcal{G}} \boldsymbol{Y} \mid \boldsymbol{Z}$.

When the graph is clear from context, we will write $\perp$ in place of $\perp_{\mathcal{G}}$. When sets $\boldsymbol{X}, \boldsymbol{W}, \boldsymbol{Z}$ satisfy $\boldsymbol{X} \perp \boldsymbol{W} \mid \boldsymbol{Z}$ they are conditionally independent: $P(\boldsymbol{X}, \boldsymbol{W} \mid \boldsymbol{Z}) = P(\boldsymbol{X} \mid \boldsymbol{Z}) P(\boldsymbol{W} \mid \boldsymbol{Z})$ (Verma & Pearl, 1988).

If we know that a deterministic mixed policy is being followed, then we may deduce further conditional independence relations. This is because conditioning on variables $\boldsymbol{V}$ may determine some decision variables, which are called "implied" (Lee & Bareinboim, 2020), or "functionally determined" (Geiger & Pearl, 1990), making them conditionally independent of other variables in the graph.

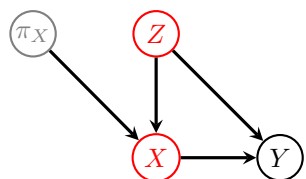

**Figure 2:** A graph where decisions $Z, X$ jointly determine the outcome $Y$. A policy node $\pi_X$ is shown, which decides the decision rule at $X$.

**Definition 6** (Implied variables; Lee & Bareinboim, 2020)**.** To obtain the *implied variables* $\lceil \boldsymbol{Z} \rceil$ for variables $\boldsymbol{Z}$ in $\mathcal{G}$ given a mixed policy scope $\mathcal{S}$, begin with $\lceil \boldsymbol{Z} \rceil \leftarrow \boldsymbol{Z}$, then add to $\lceil \boldsymbol{Z} \rceil$ every decision $X$ such that $\boldsymbol{C}_X \subseteq \lceil \boldsymbol{Z} \rceil$, until convergence.

For example, in Figure 2, we see that $\lceil X \rceil = \{Z, X\}$, so $Z$ is d-separated from $Y$ given $\lceil X \rceil$. This means that under a deterministic mixed policy, $Z$ and $Y$ are statistically independent given $X$. This has implications for materiality. In particular, it means that the best deterministic mixed policy $Z = z, X = x$ does not need to observe $Z$ at $X$. Moreover, the performance of the best deterministic mixed policy can never be surpassed by a stochastic policy (Lee & Bareinboim, 2020, Proposition 1), so $Z$ is immaterial.

## 3 Literature review

Our review will begin with the origin and applications of the concept of materiality, then cover graphical criteria in single-decision, soluble, and general multi-decision settings.

### 3.1 Graphical criteria for materiality, and their applications

"Value of information" was originally described separately from graphical models like the influence diagram (Howard, 1966). When influence diagrams were developed, however, knowing the value of information of different variables became a fundamental aspect of understanding an influence diagram (Shachter, 1986; Matheson, 1990; Howard & Matheson, 2005). Finding that a variable has zero value of information is especially important because it narrows the search for an optimal policy. The property of having zero value of information has since been termed "immateriality" (Shachter, 2016).

Various past works have sought to establish the circumstances in which a variable could be proved immaterial. Sound criteria have been proposed by Fagiuoli & Zaffalon (1998); Lauritzen & Nilsson (2001); Shachter (2016). Furthermore, Fagiuoli & Zaffalon (1998) attempted to prove the criterion's completeness, although it was not successful (Everitt et al., 2021).

There are also criteria that establish whether a variable may be valuable to control (Fagiuoli & Zaffalon, 1998; Shachter & Heckerman, 2010). In particular, establishing immateriality can help with establishing zero value of control because it shows that there exists an alternative influence diagram, where the same utility can be achieved but with fewer edges, making it easier to establish that some variables are unnecessary to control (Fagiuoli & Zaffalon, 1998). For example, in the influence diagram $Z \to X \to Y$, we know that $Z$ is immaterial, and the graph may be separated into $Z; X \to Y$, wherein $Z$ affects nothing, and so is clearly of no value to control.

In recent years, influence diagrams have been applied to evaluating the safety of AI systems. To achieve this, incentive concepts have been developed that are related to the value of information and control, such as *instrumental control incentives* (Everitt et al., 2021), *response incentives* (Everitt et al., 2021), and notions of *intent* (Halpern & Kleiman-Weiner, 2018). These concepts and their graphical criteria have been used to analyse agent interactions, especially in the context of AI, including matters of fairness (Ashurst et al., 2022), manipulation (Ward et al., 2024b), honesty (Ward et al., 2024a), and human control (Carey & Everitt, 2023). For these incentive concepts, proofs of the soundness and completeness of their graphical criteria directly extend the proofs pertaining to materiality (Everitt et al., 2021), and thus a complete criterion for materiality is a key step for this line of work.

### 3.2 Single-decision settings

In the single-decision setting, there is a sound and complete criterion for materiality: in a scoped graph $\mathcal{G}(\mathcal{S})$, there exists an SCM where the context $Z \in \boldsymbol{C}_X$ is material if and only if $Z \not\perp Y \mid \boldsymbol{C}_X \cup \{X\} \setminus \{Z\}$ and the outcome $Y$ is a descendant of $X$ (Lee & Bareinboim, 2020; Everitt et al., 2021). The proofs of the soundness (the *only if* direction) and completeness (the *if* direction) are both relevant to the current paper.

The argument in the *only if* direction is that if $X$ is not an ancestor of the outcome $Y$, then its policy is completely irrelevant to the expected utility, and so all of its contexts are immaterial, and if $Z$ is conditionally independent of the outcome $Y$ given the decision and other observations, then it may be safely ignored without changing the expected utility of the policy. These conditions imply immateriality in the multi-decision setting for precisely the same reasons.

The *if* direction is more important to us, because the result is extended by the present work. The if direction is proved by constructing a decision problem where $Z$ is material. By assumption, there is a directed path $X \dashrightarrow Y$, called the *control path*, and a path $Z \dashdash Y$, active given $\boldsymbol{C}_X \cup \{X\} \setminus \{Z\}$, called the *info path*. The idea is to construct an SCM where the info path makes $Z$ contain information about $Y$, and how $X$ ought to influence $Y$, while the control path gives $X$ a way to exert that influence.

The construction from Everitt et al. (2021) and Lee & Bareinboim (2020) has two cases. When the info path contains no colliders, the construction is shown in Figure 3a. In essence, the info path transmits a random

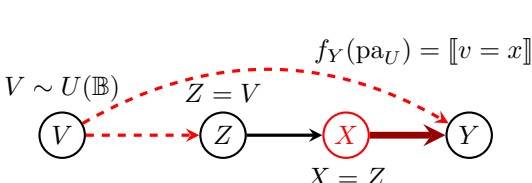

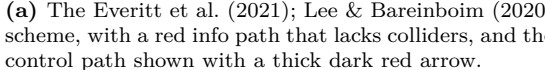

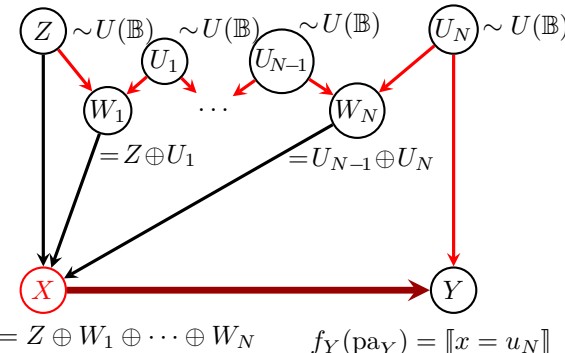

**(a)** The Everitt et al. (2021); Lee & Bareinboim (2020) scheme, with a red info path that lacks colliders, and the control path shown with a thick dark red arrow.

**(b)** The Everitt et al. (2021); Lee & Bareinboim (2020) scheme, with a red info path that contains colliders, and the control path shown in dark red.

**Figure 3:** Two decision problems where $Z$ is material for $X$. For readability, we marginalize out exogenous variables from the SCM, so $z \sim U(\mathbb{B})$ can be understood as shorthand for $z = \varepsilon_Z$ where $\varepsilon_Z \sim U(\mathbb{B})$, and so on.

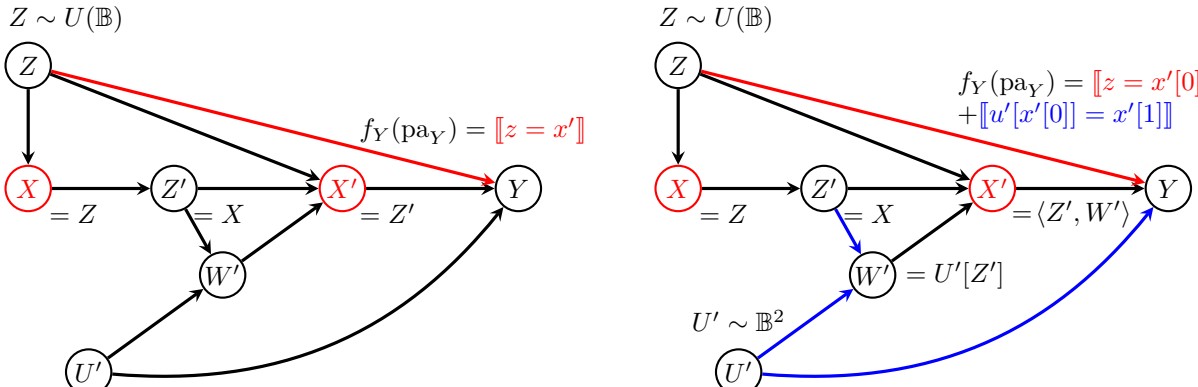

**(a)** The Everitt et al. (2021) scheme is applied using just the red info path; $Z$ is immaterial for $X$.

**(b)** The van Merwijk et al. (2022) scheme is applied, using the red and blue info paths; $Z$ is material for $X$.

**Figure 4:** Two decision problems on a soluble graph.

value $V = v$ to $\mathrm{Pa}_Y$, and the decision has to match this value, i.e. $X = V$, in order to obtain the maximum utility of $Y = 1$. Without the context $Z$, the maximum expected utility is 0.5, proving materiality.[5]

For the case where the info path does contains a collider, the construction is shown in Figure 3b. Each fork $U_i$ in the info path, along with $Z$, generates a random bit, while each collider $W_i$ is assigned the XOR ($U_{i-1} \oplus U_i$) of its two parents. By observing $z$ and the values $\boldsymbol{w}_{1:N}$, the agent has just enough information to recover $u_N$. In particular, the policy that sets $X$ equal to the XOR of $z$ and $\boldsymbol{w}_{1:N}$, obtains $X = U_N$ surely and achieves the MEU, $\mathbb{E}[Y] = 1$. Without the context $Z$, the MEU becomes 0.5, so $Z$ is material.

### 3.3 Soluble multi-decision settings

A decision problem is said to be soluble (or to have "sufficient recall") if the decision-maker always remembers enough aspects of past decisions and observations to make the present decision optimally. Formally, this requires the graph to admit an ordering $\prec = \langle X_1, \ldots, X_N \rangle$ over decisions such that for every $X_i$, for every previous decision or context $V \in \{X_j \cup C_{X_j} \mid j \prec i\}$, we have $V \notin \mathrm{Anc}(Y)$ or $V \perp Y \mid \{X_i\} \cup C_{X_i}$. For

---

[5]To be precise, the formalism of Lee & Bareinboim (2020) also allows the active path from $Z$ to include one or more bidirected edges $V \leftrightarrow Y$, but to deal with these cases, we begin with the distribution that we would use for a path $V \leftarrow L \rightarrow Y$, then marginalize out $L$.

example, in Figure 4a, using the ordering $X \prec X'$, the nodes $Z, X$ are d-separated from $Y$ by $X'$ and its contexts $\{Z, Z', W\}$, which implies solubility.

For those graphs that satisfy this condition (corresponding to solubility), there exists a sound and complete criterion for evaluating materiality. As in the single-decision setting, an observation $Z$ is identified as immaterial if $Z \perp Y \mid \boldsymbol{C}_X \cup \{X\} \setminus \{Z\}$ where $Y \in \text{Desc} X$. The only difference is that in the multi-decision setting, we can remove $Z$ as a context of $X$ and then repeat the process (i.e. the edge $Z \to X$ is removed from the scoped graph $\mathcal{G}_{\mathcal{S}}$) to find more immaterial contexts, until all immaterial observations have been removed. Conversely, when every such context has been removed in this way, then all remaining contexts must be material. For example, in the graph of Figure 4a, every decision is an ancestor of $Y$, and every context has an info path (the info paths include $Z \to Y$, $Z' \to W' \leftarrow U' \to Y$, and $W' \leftarrow U' \to Y$), so, all contexts may be material in at least one decision problem with this causal structure, and in fact they are material in the SCM shown.

It is useful to understand how the completeness direction of this result is proved, i.e. how the presence of info paths is used to construct a decision problem where $Z$ is material (van Merwijk et al., 2022, Theorem 7). We can notice what obstacles arise by considering the graph from Figure 4. If we were to apply the single-decision construction of Everitt et al. (2021) to this graph, we would first identify the info path $Z \to Y$ and the control path $X \to Z' \to X' \to Y$. The info path has no colliders, so we would construct a decision problem using the scheme from Figure 3a, and the result is shown in Figure 4a. The idea of this construction is that $X$ should have to copy $Z$ in order for the value $z$ transmitted by the info path to match the value $x'$ transmitted by the control path. Whatever action $x$ is selected, however, the decision $X'$ can assume the value $z$, thereby achieving the MEU. The MEU is then achievable whether $Z$ is a context of $X$ or not, so $Z$ is immaterial in this construction.

In order to render $Z$ material, van Merwijk et al. (2022) instead adapts the construction from Figure 4a by incentivizing $X'$ to pass along the value of $Z'$. This is done using the second info path $Z' \to W' \leftarrow U' \to Y$, shown in Figure 4b. The term $y_2 := [\![u'[x'[0]] = x'[1]]\!]$ is added to the reward, which equals 1 if $X'$ presents one bit from $U'$, along with its index. (Recall that $x'[0]$ denotes the $0^{\text{th}}$ bit of $x'$ and so on.) Furthermore, $W'$ is defined as $W' = U'[Z']$, so that $X'$ knows only the $Z'^{\text{th}}$ bit of $U'$, and since the index $z'$ is one bit, $U'$ is defined as two bits in length, i.e. $U' \sim U(\mathbb{B}^2)$. Finally, rather than requiring $z = x'$ as in Figure 4a, the term $y_1 := [\![z = x'[0]]\!]$ is included, because an optimal policy in the new model will have $x'[0] = z' = z$ (rather than simply $x' = z$, as in the previous model). In the resulting non-intervened model, the utility is clearly $Y = 2$, which is the MEU, and to achieve this utility, it is necessary that $Y_1 = Y_2 = 1$ with probability 1. To maximize $y_2$, the decision $X'$ must reproduce the only known digit from $U'$, i.e. $x' = \langle z', u'[z'] \rangle$. To maximize $y_1$, we must have $Z = X'[0]$ almost surely, and since $X'[0] = X$, this requires $X = Z$ with probability 1. This can only be done if $Z$ is a context of $X$, meaning that $Z$ is material for $X$. A key idea of this approach is that if a control path for $X$, such as $X \to Z' \to X' \to Y$, contains decisions other than $X$, then we need to incentivize the downstream decision to copy information along the control path, and this will be done by choosing values for variables lying on the info path for $X'$ (the one shown in blue in Figure 4b); we will reuse this idea in the proofs of our main result.

### 3.4 Multi-decision settings in full generality

For insoluble decision problems, the literature includes a graphical condition for establishing immateriality, but we do not yet know whether it is complete (in that materiality is possible whenever it is not satisfied).

For this procedure, we would begin with graphical criteria from the single decision case.

- If a decision $X$ is a non-ancestor of $Y$, then its contexts are immaterial,
- If $C \perp Y \mid \boldsymbol{C}_X \setminus \{C\}$, then the context $C$ is immaterial.

If either of these conditions describes the context $Z$ that we are interested in, then we have proved it immaterial, and our job is done. If neither of these conditions holds, then we can apply the more sophisticated criterion of Lee & Bareinboim (2020, Lemma 1) and Lee & Bareinboim (2020, Theorem 2). If the assumptions of Lee & Bareinboim (2020, Lemma 1) hold for some target variables $\boldsymbol{Z}$, target actions $\boldsymbol{X}'$, and latent

variables $\boldsymbol{U}'$, then they admit a factorization, which we term "LB-factorizability", after the authors' initials. Lee & Bareinboim (2020, Theorem 2) then contains some further assumptions; if these also hold, then the contexts $\boldsymbol{Z}$ are immaterial to the decisions $\boldsymbol{X}'$. We begin with the definition of LB-factorizability.

**Definition 7.** For a scoped graph $\mathcal{G}_{\mathcal{S}}$, we will say that target actions $\boldsymbol{X}'$, endogenous variables $\boldsymbol{Z}$ disjoint with $\boldsymbol{X}'$, contexts $\boldsymbol{C}' := \boldsymbol{C}_{\boldsymbol{X}'} \setminus (\boldsymbol{X}' \cup \boldsymbol{Z})$ and exogenous variables $\boldsymbol{U}'$ are *LB-factorizable* if there exists an ordering $\prec$ over $\boldsymbol{V}' := \boldsymbol{C}' \cup \boldsymbol{X}' \cup \boldsymbol{Z}$ such that:

I. $(Y \perp \boldsymbol{\pi}_{\boldsymbol{X}'} \mid \lceil (\boldsymbol{X}' \cup \boldsymbol{C}') \rceil)$,

II. $(C \perp \boldsymbol{\pi}_{\boldsymbol{X}'_{\prec C}}, \boldsymbol{Z}_{\prec C}, \boldsymbol{U}' \mid \lceil (\boldsymbol{X}' \cup \boldsymbol{C}')_{\prec C} \rceil)$, for every $C \in \boldsymbol{C}'$ and

III. $\boldsymbol{V}'_{\prec X}$ is disjoint with $\mathrm{Desc}(X)$ and subsumes $\mathrm{Pa}(X)$ for every $X \in \boldsymbol{X}'$,

where $\boldsymbol{\pi}_{\boldsymbol{X}'}$ consists of a new parent $\pi_X$ added to each variable $X \in \boldsymbol{X}'$,[6] and $\boldsymbol{W}_{\prec V}$, for $\boldsymbol{W} \subseteq \boldsymbol{V}'$, denotes the subset of $\boldsymbol{W}$ that is strictly prior to $V$ in the ordering $\prec$.

For example, consider the graph Figure 2. In this case, $Y \in \mathrm{Desc}(X)$ and $Z \not\perp Y \mid X$, so the single-decision criteria cannot establish that $Z$ is immaterial for $X$. However, by choosing $\boldsymbol{Z} = \{Z\}$, $\boldsymbol{X}' = \{X\}$, and the ordering $\prec = \langle Z, X \rangle$, we have that:

I. the outcome $Y$ is d-separated from $\boldsymbol{\pi}_X$ by $\lceil X \rceil$, (since $Z$ is a decision that lacks parents, we actually have $\lceil X \rceil = \{Z, X\}$),

II. the contexts $\boldsymbol{C}'$ are an empty set, so (II) is trivially true, and

III. $\boldsymbol{V}'_{\prec X} = \boldsymbol{Z}$, and $\boldsymbol{Z}$ is disjoint with $\mathrm{Desc}(X)$ and $\boldsymbol{Z} \supseteq \mathrm{Pa}(X)$

so $\boldsymbol{Z}$ and $\boldsymbol{X}'$ are LB-factorizable.

In this paper, our focus is exclusively on the assumptions of Lee & Bareinboim (2020, Lemma 1) rather than the additional conditions of Lee & Bareinboim (2020, Theorem 2), but for completeness sake, the latter is reproduced in Section A. We also establish in Section A that the assumptions of Lee & Bareinboim (2020, Theorem 2) are indeed satisfied in the graph Figure 2, meaning that $Z$ is immaterial for $X$, matching the ad hoc analysis of this graph in Section 2.

## 4 Main result

### 4.1 Theorem statement and proof overview

The goal of this paper is to prove that condition (I) of LB-factorizability is necessary to establish immateriality. More precisely, if condition (I) is unsatisfiable for all observations in the graph, then the graph is compatible with materiality for all observations. It might initially seem unnecessarily stringent to assume that this holds for *all* observations, rather than the context $Z_0$ for which we are trying to prove materiality. Recall from Figure 4b, however, that proofs of materiality are recursive, in that to prove that $Z$ is material for $X$, we incentivized $X$ to copy $Z$, and to do this, we had to incentivize $X'$ to pass on the value of $Z'$. This required us to assume that other contexts and decisions (such as $Z'$ and $X'$) have their own info paths and control paths, not just $Z$ and $X$. So that we can assume this, in our theorem below, condition (C) requires that (I) holds for all contexts. Conditions (A) and (B) are also necessary for a graph to be compatible with materiality, because their negation implies immateriality, as discussed in Section 3.2.

**Theorem 8.** *If, in a scoped graph $\mathcal{G}_{\mathcal{S}}$, for every $X \in \boldsymbol{X}(\mathcal{S})$*

A. $X \in Anc_{\mathcal{G}_{\mathcal{S}}}(Y)$,

B. $\forall C \in \boldsymbol{C}_X : (C \not\perp_{\mathcal{G}_{\mathcal{S}}} Y \mid (\{X\} \cup \boldsymbol{C}_X \setminus \{C\}))$, *and*

---

[6]To be precise, each d-separation $\perp$ in (A-B) holds in the graph $\mathcal{G}'$, obtained from $\mathcal{G}$ by adding a parent $\pi_X$ for each decision $X$.

C. *for every context $Z \in \boldsymbol{C}_X$ in $\mathcal{G}_{\mathcal{S}}$, $(\pi_X \not\perp_{\mathcal{G}_{\mathcal{S}}} Y \mid \lceil(\boldsymbol{X}(\mathcal{S}) \cup \boldsymbol{C}_{\boldsymbol{X}(\mathcal{S}) \setminus \{Z\}}) \setminus \{Z\}\rceil)$, where $\pi_X$ is a new parent of $X$,*

*then for every $X_0 \in \boldsymbol{X}(\mathcal{S})$ and $Z_0 \in \boldsymbol{C}_{X_0}$, there exists an SCM where $Z_0$ is material for $X_0$.*

In words, this means that if an outcome $Y$ is influencable (condition A) and each context provides information about $Y$ given other contexts (condition B), and given everything determined by the contexts and decisions (condition C), then each context is material in at least one model compatible with the graph.

We will prove this result in three stages, across the next three sections.

- In Section 4.2, we will prove that any scoped graph satisfying the conditions of Theorem 8 contains certain paths called "materiality paths", which begin at a context $Z_0$ and decision $X_0$.
- In Section 4.3, we will use the materiality paths to define a model for this scoped graph, which we will call the *materiality SCM*.
- In Section 4.4, we will prove that in the materiality SCM, $Z_0$ is material for $X_0$.

## 4.2 The materiality paths

To prove materiality, we will begin by selecting info paths and a control path, similar to what was described in Section 3.3 and illustrated in Figure 4b. One difference, however, is that we must choose paths that allow us to prove the value of remembering a past decision. We will describe how to accommodate this difference in Section 4.2.1 then define the paths in Section 4.2.2.

### 4.2.1 Paths for the value of remembering a decision

Suppose that given the graph in Figure 5, we would like to establish that it is useful to remember the decision $Z_0$, when making the decision $X_0$. Given an info path, the procedures of Everitt et al. (2021) and van Merwijk et al. (2022) will tell us how to construct an SCM, in which we can compute whether $Z_0$ is material. The problem is that there are two candidate info paths: $Z_0 \to Y$ and $Z_0 \leftarrow U \to Y$, and these procedures give us no guidance on which one to choose. So which one will lead us to construct an SCM where $Z_0$ is material? Let us consider each info path in turn.

Using the info path $Z_0 \to Y$, we obtain the model in Figure 5a. In this model, we have $Y = 1$ if $x_0 = z_0$, i.e. the decision $X_0$ is required to match the value of a past decision Figure 5a. Then, the MEU of 1 can be achieved with a deterministic policy such as $Z_0 = 1, X_0 = 1$, and $Z_0$ is immaterial for $X_0$. To understand this in terms of the paths involved, Why does the proof fail with this info path? It is because the the info path $Z_0 \to Y$ includes no parents of $Z_0$, so $Z_0$ is *implied* by values outside the info path, and $Z_0 \to Y$ is rendered inactive given $\lceil U \rceil$. This means that observing $Z_0$ can no longer provide useful information about how to maximize $Y$.

Using the info path $Z_0 \leftarrow U \to Y$, we obtain the model in Figure 5b. In this model, $Y = 1$ if $x_0 = u$, i.e. the decision $X_0$ must match the value of a random Bernoulli variable $U$. The value of $U$ is directly observed only by $Z_0$, and so in an optimal policy $X_0$ must observe the decision $z_0$; this is the case in the policy $z_0 = u, x_0 = z_0$, and so $Z_0$ is material for $X_0$. The proof has succeeded because the info path $Z_0 \leftarrow U \to Y$ includes a parent $U$ of $Z_0$, meaning that $Z_0$ is no longer *implied* by variables outside the info path, and the path $Z_0 \leftarrow U \to Y$ remains active given $\lceil \emptyset \rceil$. This choice of path helps to ensure that $Z_0$ provides useful information about $Y$.

In general, when an info path passes through a decision, such as $Z_0$ in the previous example, we want it to also pass through a non-decision parent of $Z_0$. We will now see that condition (C) of Theorem 8 implies that each decision has a non-decision parent $N$ that is not in $\lceil(\boldsymbol{X}(\mathcal{S}) \cup C_{\boldsymbol{X}(\mathcal{S}) \setminus \{Z\}}) \setminus \{Z\}\rceil$.

**Lemma 9.** *If a scoped graph $\mathcal{G}(\mathcal{S})$ satisfies condition (C) of Theorem 8, then for every context $Z \in \boldsymbol{C}_X$ where $Z, X \in \boldsymbol{X}(\mathcal{S})$ are decisions, there exists a non-decision $N \in \boldsymbol{C}_Z \setminus \lceil(\boldsymbol{X}(\mathcal{S}) \cup C_{\boldsymbol{X}(\mathcal{S}) \setminus \{Z\}}) \setminus \{Z\}\rceil$.*

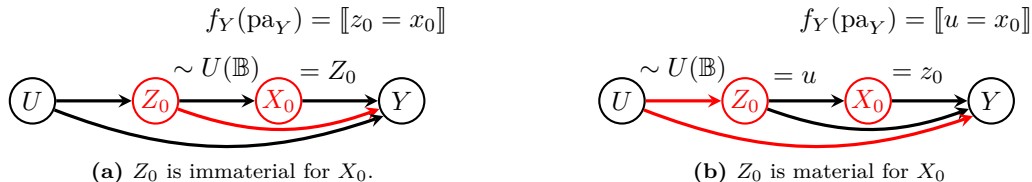

**(a)** $Z_0$ is immaterial for $X_0$.

**(b)** $Z_0$ is material for $X_0$

**Figure 5:** Two SCMs, with models constructed using different (red) info paths.

Intuitively, this is because condition (C) states that there is an active path from $Z$ to $Y$, given a superset of $\lceil \boldsymbol{X}(\mathcal{S}) \setminus \{Z\} \rceil$. If all of the parents of $Z$ are decisions, then we would have $Z \in \lceil \boldsymbol{X}(\mathcal{S}) \setminus \{Z\} \rceil$, and every path would be blocked, and condition (C) could not be true.

*Proof of Lemma 9.* Assume that there is no such non-decision $N$, i.e. $\boldsymbol{C}_Z \subseteq \lceil (\boldsymbol{X}(\mathcal{S}) \cup C_{\boldsymbol{X}(\mathcal{S})\setminus\{Z\}}) \setminus \{Z\} \rceil$, and that $\pi_X \not\perp Y \mid \lceil (\boldsymbol{X}(\mathcal{S}) \cup C_{\boldsymbol{X}(\mathcal{S})\setminus\{Z\}}) \setminus \{Z\} \rceil$, (by condition (C) of Theorem 8), and we will prove a contradiction. From $\boldsymbol{C}_Z \subseteq \lceil (\boldsymbol{X}(\mathcal{S}) \cup C_{\boldsymbol{X}(\mathcal{S})\setminus\{Z\}}) \setminus \{Z\} \rceil$, we deduce that $Z \in \lceil (\boldsymbol{X}(\mathcal{S}) \cup C_{\boldsymbol{X}(\mathcal{S})\setminus\{Z\}}) \setminus \{Z\} \rceil$ (by the definition of $\lceil \boldsymbol{W} \rceil$), and then there can be no active path from $\pi_X$ to $Y$ given $\lceil (\boldsymbol{X}(\mathcal{S}) \cup C_{\boldsymbol{X}(\mathcal{S})\setminus\{Z\}}) \setminus \{Z\} \rceil \supseteq \boldsymbol{C}_Z \cup \{Z\}$, contradicting condition (C) of Theorem 8, and proving the result. $\square$

This tells us that for any decision $Z$ there is an edge $Z \leftarrow N$. Moreover, by condition (C) of the main result, we know that there is an info path from $N$ to $Y$. By concatenating the edge and the path, we can obtain a path from $Z$ to $Y$, which we will prove is active given $\lceil (\boldsymbol{X}(\mathcal{S}) \cup C_{\boldsymbol{X}(\mathcal{S})\setminus\{Z\}}) \setminus \{Z\} \rceil$. This is precisely what we want for our info path, first because it passes through a non-decision parent of the endpoint $Z$, (so the endpoint $U$ will not be determined as it is in Figure 5a) and second, because this path is active given $\lceil (\boldsymbol{X}(\mathcal{S}) \cup C_{\boldsymbol{X}(\mathcal{S})\setminus\{Z\}}) \setminus \{Z\} \rceil$, meaning that forks and chains are not decisions, and so they cannot be determined either.

**Lemma 10.** *If a scoped graph $\mathcal{G}(\mathcal{S})$ satisfies conditions (B-C) of Theorem 8, then for every edge $Z \rightarrow X$ between decisions $Z, X \in \boldsymbol{X}(\mathcal{S})$, there exists a path $Z \leftarrow N \dashrightarrow Y$, active given $\lceil (\boldsymbol{X}(\mathcal{S}) \cup C_{\boldsymbol{X}(\mathcal{S})\setminus\{Z\}}) \setminus \{Z\} \rceil$, (so $N \notin \lceil (\boldsymbol{X}(\mathcal{S}) \cup C_{\boldsymbol{X}(\mathcal{S})\setminus\{Z\}}) \setminus \{Z\} \rceil$).*

The proof that the segment $N \dashrightarrow Y$ is active given $\lceil (\boldsymbol{X}(\mathcal{S}) \cup C_{\boldsymbol{X}(\mathcal{S})\setminus\{Z\}}) \setminus \{Z\} \rceil$ rather than just $\lceil (\boldsymbol{X}(\mathcal{S}) \cup C_{\boldsymbol{X}(\mathcal{S})\setminus\{N\}}) \setminus \{N\} \rceil$ requires some detail, so it is deferred to Section B.1.

### 4.2.2 Defining the materiality paths

In this subsection, we will describe how to select some paths to exhibit that some context $Z_0$ is material for a decision $X_0$. In overview, this will begin with the selection of a directed control path passing through $Z_0 \rightarrow X_0$ and terminating at $Y$. There are finitely many info paths, which emanate from the control paths and go to $Y$. Then finally, there are finitely many auxiliary paths, which go from colliders in the info paths to $Y$. We will refer to the control, info, and auxiliary paths collectively as the *materiality paths*, and they can all be seen in Figure 6.

The control path is the horizontal spine in Figure 6. The control path exists because $X_0$ is an ancestor of $Y$ by condition (A) of Theorem 8, and $X_0$ has a chance node ancestor $A$ by Lemma 10 (because the parent $Z_0$ of $X_0$ is either a chance node, or it has a chance node parent.)

We then index the decisions on the control path as $X_{i_{\min}}, \ldots, X_{i_{\max}}$, and their parents along the control path as $Z_{i_{\min}}, \ldots, Z_{i_{\max}}$. where $i_{\min}$ is either $0$ (if $Z_0$ is a chance node), or $-1$ (if $Z_0 = X_{-1}$). (Note that where the control path has consecutive decisions, we will have $Z_i = X_{i-1}$.) Then, an info path $m'_i$ is chosen for each context $Z_i$ using Lemma 9 to ensure that it satisfies some desirable properties.

Rather than the info path $m'_i$ per se, we will often be interested in the portion of it that is non-overlapping with the control path. So the *intersection node $T_i$* will essentially be the point at which the info path departs from the control path, and the truncated info path $m'_i$ will be the segment $Y_i \dashrightarrow Y$ of $m_i$.

Then, in each truncated info path $m_i$, for each collider $W_{i,j}$, we will define an *auxiliary path $r_{i,j} : W_{i,j} \dashrightarrow Y$*.

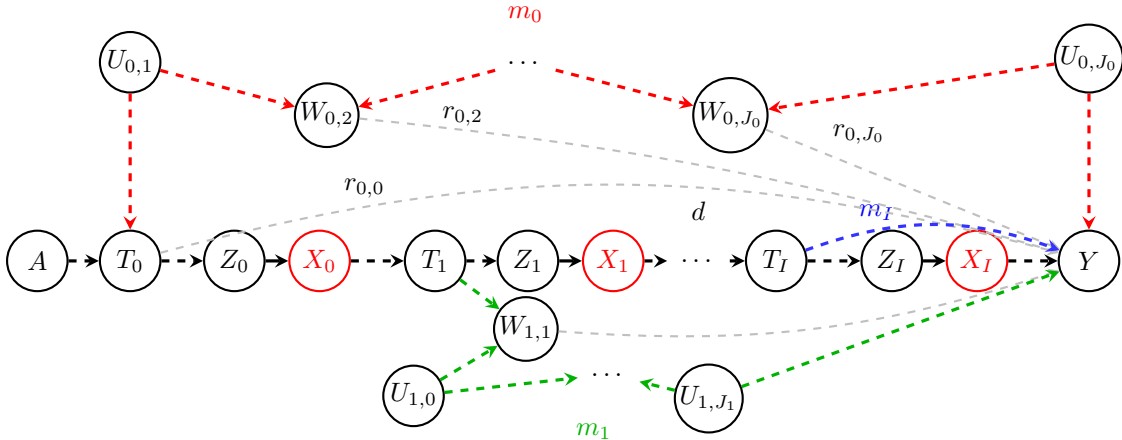

**Figure 6:** The set of paths proven to exist by Lemma 11 are red, green and blue. In each case, the point of departure of the active path from the (black) directed path is designated by $T_i$. In full generality, each path may begin either as $Z_i \dashleftarrow T_i \leftarrow \cdot$ (as in red), or as $Z_i \dashleftarrow T_i \rightarrow \cdot$ (green, blue).

**Lemma 11.** *Let $\mathcal{G}(\mathcal{S})$ be a scoped graph that contains a context $Z_0 \in \boldsymbol{C}_{X_0}$ and satisfies the conditions of Theorem 8. Then, it contains the following:*

- *A **control path**: a directed path $d : A \dashrightarrow Z_0 \rightarrow X_0 \dashrightarrow Y$, where $A$ is a non-decision, possibly equal to $Z_0$, and $d$ contains no parents of $X_0$ other than $Z_0$.*
- *We can write $d$ as $A \dashrightarrow Z_{i_{min}} \rightarrow X_{i_{min}} \dashrightarrow \cdots Z_0 \rightarrow X_0 \dashrightarrow Z_{i_{max}} \rightarrow X_{i_{max}} \dashrightarrow Y, i_{min} \leq i \leq i_{max}$, where each $Z_i$ is the parent of $X_i$ along $d$ (where $A \dashrightarrow Z_{i_{min}}$ and $X_{i-1} \dashrightarrow Z_i$ are allowed to have length $0$). Then, for each $i$, define the **info path**: $m_i' : Z_i \dashrightarrow Y$, active given $\lceil (\boldsymbol{X}(\mathcal{S}) \cup C_{\boldsymbol{X}(\mathcal{S}) \setminus Z_i}) \setminus Z_i \rceil$, that if $Z_i$ is a decision, begins as $Z_i \leftarrow N$ (so $N \in \boldsymbol{C}_{Z_i} \setminus \lceil (\boldsymbol{X}(\mathcal{S}) \cup C_{\boldsymbol{X}(\mathcal{S}) \setminus Z_i}) \setminus Z_i \rceil$.)*
- *Let $T_i$ be the node nearest $Y$ in $m_i' : Z_i \dashrightarrow Y$ (and possibly equal to $Z_i$) such that the segment $Z_i \overset{m_i'}{\dashrightarrow} T_i$ of $m_i'$ is identical to the segment $Z_i \overset{d}{\dashleftarrow} T_i$ of $d$. Then, let the **truncated info path** $m_i$ be the segment $T_i \overset{m_i'}{\dashrightarrow} Y$.*
- *Write $m_i$ as $m_i : T_i \dashrightarrow W_{i,1} \dashleftarrow U_{i,1} \dashrightarrow W_{i,2} \dashleftarrow U_{i,2} \cdots U_{i,J_i} \dashrightarrow Y$, where $J_i$ is the number of forks in $m_i$. (We allow the possibilities that $T_i = W_{i,1}$ so that $m_i$ begins as $T_i \dashleftarrow U_{i,1}$, or that $J_i = 0$ so that $m_i$ is $T_i \dashrightarrow Y$.) Then, for each $i$ and $1 \leq j \leq J_i$, let the **auxiliary path** be any directed path $r_{i,j} : W_{i,j} \dashrightarrow Y$ from $W_{i,j}$ to $Y$.*

The proof was described before the lemma statement, and is detailed in Section B.2.

## 4.3 The materiality SCM

We will now present a construction that uses the materiality paths to define an SCM where $Z_0$ is material for $X_0$. This construction will differ from those of Sections 3.2 and 3.3, in ways that help to deal with insolubility. We will outline these differences in Section 4.3.1, then define the construction in Section 4.3.2.

### 4.3.1 Models for insoluble graphs

To see how the past constructions of Everitt et al. (2021) and van Merwijk et al. (2022) need to be modified, it is instructive to consider some insoluble graphs that are allowed by Theorem 8, and examine how the past constructions fail to establish materiality.

Consider, for example, the graph of Figure 7a. This is insoluble, because at the decision $X_0$, the past decision $X'$ contains information about $Y$, i.e. $X' \not\perp Y \mid Z_0$, and it also satisfies the three conditions of Theorem 8.

In this graph, we can select the control path $X_0 \rightarrow Y$ and info path $Z_0 \rightarrow W_1 \leftarrow U_1 \rightarrow Y$. Then, the construction from Everitt et al. (2021) and van Merwijk et al. (2022) could be applied if only $X'$ was a

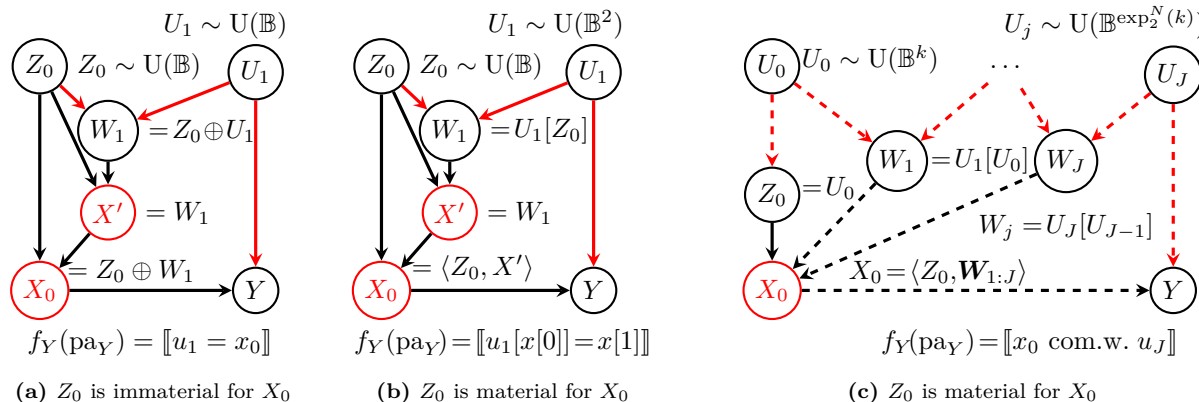

**Figure 7:** Two SCMs (a-b), and a description of a family of SCMs, where each dashed line represents a path. The repeated exponent $\exp_2^n(k)$ is defined as $k$ if $n = 0$, and $2^{\exp_2^{n-1}(k)}$ otherwise.

non-decision variable. So treat $X'$ as though it was a non-decision, and we obtain the decision problem shown in Figure 7a. In this model, the outcome $Y$ is equal to 1 when $X_0$ is equal to $U_1$. The idea of this construction is that since $W_1 = Z_0 \oplus U_q$, the MEU is achieved with the non-intervened policy $X_0 = Z_0 \oplus W_1$, which would require $X_0$ to depend on $Z_0$. In this model, however, $X'$ is a decision, and so there exists an alternative optimal policy, where $X' = U_1$ and $X_0 = X'$, and $X_0$ need not directly depend on $Z_0$, proving that $Z_0$ is immaterial for $X_0$. Essentially, the single bit of $X'$ suffices to transmit the value of $U_1$, and so the information contained in $Z_0$ was not incrementally useful. So long as the decision problem allows $X'$ to copy $U_1$, $Z_0$ will be immaterial.

A potential remedy is to give $U_1$ a larger domain than $X'$, and this is the idea behind the modified construction shown in Figure 7b. In this scheme, *two* random bits are generated at $U_1$. The outcome is $Y = 1$ if $X_1$ supplies one bit from $U_1$ along with its index. A random bit is sampled at $Z_0$, and $W_1$ presents the $Z_0^{\text{th}}$ bit from $U_1$, while $X_1$ has a domain of just one bit. Then, similar to our previous discussion of Figure 4b, the only bit from $U_1$ that $X_0$ can reliably know is the $Z_0^{\text{th}}$ bit. Hence the only way to achieve the MEU is for $X'$ to inform $X_0$ about the value of $W_1$, and for $X_0$ to equal $X_0 = \langle Z_0, X' \rangle$. Importantly, this can only be done if $X_0$ observes $Z_0$, meaning that $Z_0$ is material for $X_0$.

How might we generalize this construction to the case of Figure 7c, where the info path has multiple fork variables? Instead of just $Z_0$ and $U_1$, we now have $J+1$ fork variables $\boldsymbol{U}_{0:J}$, which sample bitstrings, each one $2^n$ as long as its predecessor. Each bitstring is then used to index its successor to obtain one bit. Formally, the observation is $Z_0 = U_0$, and at each collider $W_i$, we have $W_i = U_j[U_{j-1}]$. If the decision $X_0$ copies the values from $Z_0$ and $\boldsymbol{W}_{1:J}$, we will say that it is "consistent" with $\boldsymbol{u}_{0:J}$. The outcome $Y$ in this graph must be defined in terms of only $X_0$ and $U_J$, so it cannot verify whether $X_0 = \boldsymbol{U}_{0:J}$. What it can do, however, is check whether there exists *any* assignments $\boldsymbol{U}_{0:J-1}$ that, taken with $U_J$ would produce assignments $z_0, \boldsymbol{w}_{1:J}$ that $X_0$ is consistent with. We term this latter property "compatibility".

**Definition 12** (Consistency and compatibility). Let $\boldsymbol{w} = \langle w_0, w_1, \ldots, w_J \rangle$ where $w_0 \in \mathbb{B}^k$ and $w_n \in \mathbb{B}$ for $n \geq 1$. Then, $\boldsymbol{w}$ is *consistent with* $\boldsymbol{u} = \langle u_0, \ldots, u_J, u_i \in \mathbb{B}^{\exp_2^i(k)} \rangle$ (i.e., $\boldsymbol{w} \sim \boldsymbol{u}$) if $w_0 = u_0$ and $w_n = u_n[u_{n-1}]$ for $n \geq 1$. Moreover, $\boldsymbol{w}$ is *compatible with* $u_J \in \mathbb{B}^{\exp_2^J(k)}$ (i.e. $\boldsymbol{w} \sim u_J$) if there exists any $u_0, \ldots, u_{J-1}$ such that $\boldsymbol{w}$ is consistent with $u_0, \ldots, u_J$.

For example, in Figure 7b, $x_1$ is consistent with $\langle z_0, u_1 \rangle$ if $x_1$ produces the $z_0^{\text{th}}$ bit from $u_1$, i.e. $x_1 = \langle z_0, u_1[z_0] \rangle$. More leniently, we will say that $x_1$ is *compatible* with $\langle z_0, u_1 \rangle$ if $x_1$ produces *any* bit from $u_1$, i.e. $x_1 = \langle z_0, u_1[b] \rangle$ for any $b$. This means that either $\langle 0, 0 \rangle$ or $\langle 1, 1 \rangle$ is compatible with $z_0 = 0$ and $u_1 = 01$, but only the former is consistent with $z_0 = 0$ and $u_1 = 00$.

The idea of the construction is that if the decision $x_0$ is inconsistent with the fork assignments, then it will also sometimes be incompatible with $U_J$, and therefore will be suboptimal. For example, in Figure 7b, suppose that we have assignments $z_0 = 0$ and $u_1 = 01$ and a decision $x = \langle 1, 1 \rangle$. Then, given the assignments

$y_0 = 0$ and $u_1 = 00$, we will have assignments $z_0 = 0$ and $w_1 = 0$, which will in turn cause the assignment $x = \langle 1, 1 \rangle$ however, this is not consistent with $z_0 = 0$ and $u_1 = \langle 0, 0 \rangle$, so the utility will be $y = 0$. We now prove that this is also true for the more general construction of Figure 7c: if with strictly positive probability, the assignment of $X_0$ is inconsistent with $\boldsymbol{u}_{0:J}$, then there will exist an alternative assignment $\boldsymbol{U}_{0:J} = \boldsymbol{u}'_{0:J}$, that produces the same assignments to the observations of $X_0$, but where $X_0$ is incompatible with $\boldsymbol{u}'_J$.

**Lemma 13.** *Let $\boldsymbol{w} = \langle w_0, \ldots, w_J \rangle$ and $\bar{\boldsymbol{w}} = \langle \bar{w}_0, \ldots, \bar{w}_J \rangle$ be sequences with $w_0, \bar{w}_0 \in \mathbb{B}^k$, $w_j, \bar{w}_j \in \mathbb{B}$ for $j \geq 1$, and let $J' \leq J$ be the smallest integer such that $w_{J'} \neq \bar{w}_{J'}$. Let $u_0, \ldots, u_{J'}$ be a sequence where $u_j[u_{j-1}] = w_j$ for $1 \leq j < J'$. Then, there exists some $u_{J'+1}, \ldots, u_J$ such that $\boldsymbol{w}$ is consistent with $u_0, \ldots, u_J$, but $\bar{\boldsymbol{w}}$ is incompatible with $u_J$.*

The proof is deferred to Section B.5.

We can then use this result to argue that $Z_0$ is material in Figure 7c. If $Z_0$ is an observation of $X_0$, then we achieve the MEU of 1 in the non-intervened model, where $x_0$ is compatible with $\boldsymbol{u}_{0:J}$, and $y = 1$ surely. If instead $Z_0$ is removed as an observation, then $X_0$ cannot copy it, because $Z_0$ is constructed to contain $k$ bits, where $k$ may be an arbitrarily large integer. It follows that $x_0$ is sometimes inconsistent with $\boldsymbol{u}_{0:J}$, and hence sometimes compatible with $u_J$ (by Lemma 13), so we will have $y = 0$ sometimes (by the definition of $Y$), making the MEU less than 1. This argument will be fully formalized in Section 4.4.

### 4.3.2 A decision problem for any graph containing the materiality paths

We will now define a construction for any graph containing the materiality paths described in Lemma 11.

In most respects, the construction is a slight evolution of previous examples. Where the truncated info path $m_i$ is a directed path, the construction will generalize Figure 3a, and the decision $X_i$ will be tasked with copying the value at an intersection node $T_i$. Where $m_i$ is not a directed path, the construction will generalize Figure 7c, and $X_i$ will be tasked with producing a bitstring that is compatible with forks $U_{i,1:J_i}$.

For this more general construction, we do, still need to fix some notation. The first issue is that the materiality paths may overlap. To accommodate this possibility, we will define a random variable $V^p$ for each variable in a materiality path $p$, then to derive the overall materiality SCM, we will define $V$ as a Cartesian product over each $V^p$. The sole exception is the outcome variable $Y$, which will be defined by a sum over every $Y^p$. For any set of paths $\boldsymbol{p}$, we define $V^{\boldsymbol{p}} = \times_{p \in \boldsymbol{p}} V^p$.

A second issue is that variables will often depend on their parents along the same path — something that we can compactly describe as follows.

**Definition 14** (Parents along paths). When a vertex $V$ has a unique parent $\bar{V}$ along $p$, $\mathrm{Pa}(V^p) = \bar{V}^p$, and for a set of paths $\boldsymbol{p}'$, let $\mathrm{Pa}(V^{\boldsymbol{p}'}) = \times_{p \in \boldsymbol{p}'} \mathrm{Pa}(V^p)$. For a collider $V$ in a truncated info path $m_i : T_i \text{ --- } Y$, let the parent nearer $T_i$ along $m_i$ be $\mathrm{Pa}_L(V)$, and the parent nearer $Y$ be $\mathrm{Pa}_R(V)$.

For example, a non-outcome child $V$ of $A$ along the control path will be assigned $V^d = \mathrm{Pa}(V^d)$.

A third issue is that variables must sometimes depend on variables that do not lie in their paths, and this occurs in three cases:

- in the truncated info path, the child of the intersection node must depend on all components $T_i^{\boldsymbol{p}_i}$, not just $T_i^{m_i}$;
- in the auxiliary path, the second node must copy the value of the collider from the info path $W_{i,j}^{m_i}$;
- when the intersection node is a collider that is also the source node $A$ of the control path, then it must depend on not just $\mathrm{Pa}(T_i^{\boldsymbol{p}_i})$, but also the exogenous variable $\mathcal{E}_A$.

We define the following shorthand to deal with these three cases:

**Definition 15** (Out-of-path parent relations). For a truncated info path $m_i$, let:

$$\mathrm{Pa}^*(V^{m_i}) = \begin{cases} T_i^{\boldsymbol{p}_i} & \text{if } \mathrm{Pa}(V^{m_i}) = T_i^{m_i} \\ \mathrm{Pa}(V^{m_i}) & \text{otherwise} \end{cases}, \text{ and } \mathrm{Pa}_L^*(V) = \begin{cases} T_i^{\boldsymbol{p}_i} & \text{if } \mathrm{Pa}_L(V^{m_i}) = T_i^{m_i} \\ \mathrm{Pa}_L(V_l^{m_i}) & \text{otherwise} \end{cases}.$$

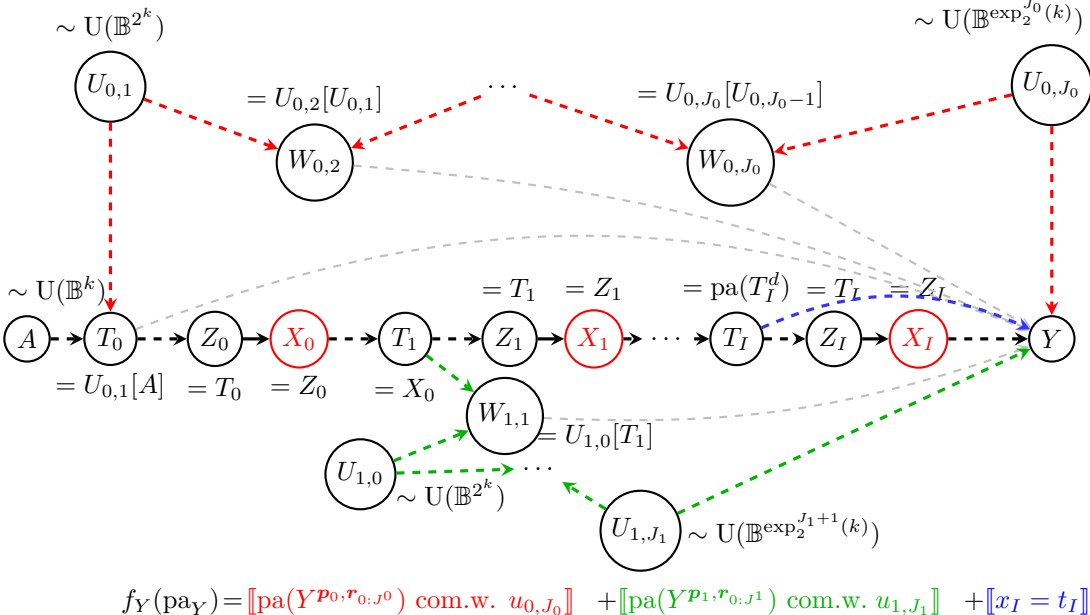

**Figure 8:** The materiality SCM: a general SCM where $Z_0$ is material for $X_0$.

For an auxiliary path $r_{i,j}$, let $\text{Pa}^*(V^{r_{i,j}}) = \begin{cases} W_{i,j}^{m_i} & \text{if } \text{Pa}(V^{r_{i,j}}) = W_{i,j}^{m_i} \\ \text{Pa}(V^{r_{i,j}}) & \text{otherwise} \end{cases}$.

For the intersection node let:

$$\text{Pa}^*(T_i^{\boldsymbol{p}_i}) = \begin{cases} \mathcal{E}_A \times \text{Pa}(T_i^{\boldsymbol{p}_i}) & \text{if } T_i \text{ is } A \\ \text{Pa}(T_i^{\boldsymbol{p}_i}) & \text{otherwise} \end{cases}.$$

Using this notation, we now define the materiality SCM as follows.

**Definition 16** (Materiality SCM)**.** Given a graph containing the materiality paths, we may define the following random variables.

In the control path, $d : A \dashrightarrow Y$, let:

- the source be $A^d = \mathcal{E}^{A^d}$ where $\mathcal{E}^{A^d} \sim \text{U}(\mathbb{B}^k)$, the positive integer $k$ is the smallest such that $2^k > (k + c)bc$, the maximum number of variables that can be contexts of one decision is $b := \max_{X \in \boldsymbol{X}(\mathcal{S})} |C_X|$, and $c$ is the maximum number of materiality paths passing through any vertex in the graph;
- every non-endpoint $V$ be $V^d = \text{Pa}(V^d)$.

In each truncated info path that is directed, $m_i : T_i \dashrightarrow Y$, let:

- the intersection node $T^{m_i}$ have trivial domain;
- each chain node be $V^{m_i} = \text{Pa}^*(V^{m_i})$;
- the outcome have the function $f_{Y^{m_i}}(\text{pa}_Y) = [\![\text{pa}(Y^{\boldsymbol{p}_i}) = \text{pa}^*(Y^{m_i})]\!]$.

In each truncated info path that is not directed, $T_i \dashrightarrow \leftarrow W_{i,1} \to \cdots \leftarrow W_{i,J} \dashrightarrow Y$, let:

- each fork be $W_{i,j}^{m_i} = \mathcal{E}^{W_{i,j}^{m_i}}$, $\mathcal{E}^{W_{i,j}^{m_i}} \sim \text{U}(\mathbb{B}^{\exp_2^j(k+|\boldsymbol{p}_i|-1)})$ where $|\boldsymbol{p}_i|$ is the number of paths in $\boldsymbol{p}_i$;
- each chain node be $V^d = \text{Pa}^*(V^d)$;

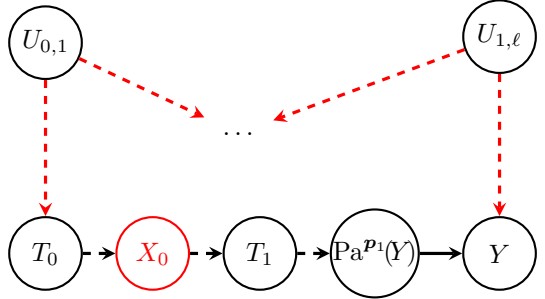
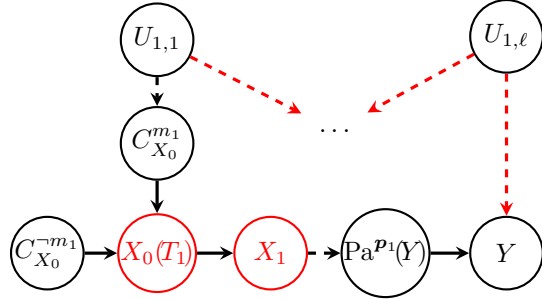

**(a)** The intersection node $T_1$ is a chance node.

**(b)** The intersection node $T_1$ is a decision. The contexts of $X_0$ are divided into $C_{X_0}^{m_1}$ (its parent along the info path), and $C_{X_0}^{\neg m_1}$ (the other parents).

**Figure 9:** The cases where the intersection node $T_1$ is a chance node, or a decision

- each collider be $V^{m_i} = \mathrm{Pa}_R(V^{m_i})[\mathrm{Pa}_L^*(V^{m_i})]$;
- each intersection node be $T_i^{m_i} = \mathrm{Pa}(V^{m_i})[\mathrm{Pa}^*(T_i^{\boldsymbol{p}_i})]$ if the info path begins as $T_i \to \cdot$, otherwise it has empty domain;
- the outcome have the function $f_{Y^{m_i}}(\mathrm{pa}_Y) = [\![\mathrm{pa}(Y^{\boldsymbol{p}_i, \boldsymbol{r}_{i,1:J_i}})$ is compatible with $\mathrm{pa}^*(Y^{m_i})]\!]$.

In each auxiliary path $r_{i,j} : W_{i,j} \to V_2 \dashrightarrow Y$, let:

- each chain node have $V^{r_{i,j}} = \mathrm{Pa}^*(v^{r_{i,j}})$.
- each source $W_{i,j}$ have trivial domain.

Then, let the *materiality SCM* have outcome variable $Y = \sum_{i_{\min} \leq i \leq i_{\max}} Y^{m_i}$, and non-outcome variables $V = \times_{p \in \{d, m_i, r_{i,1:J_i} | i_{\min} \leq i \leq i_{\max}\}} V^p$.

Note that this defines an SCM because each variable is a deterministic function of only its endogenous parents and exogenous variables.

In this non-intervened model, we have $Y^{m_i} = 1$, surely.

**Lemma 17.** *In the non-intervened model, the materiality SCM has $Y = i_{max} - i_{min} + 1$, surely.*

The proof follows from the model definition, and is supplied in Section B.4.

We also know that each utility term $Y^{m_i}$ is upper bounded at one, so in order to obtain the MEU, each $Y^i$ must equal 1, almost surely.

**Lemma 18.** *If a policy $\boldsymbol{\pi}$ for the materiality SCM, has $P^{\boldsymbol{\pi}}(Y^{m_i} < 1) > 0$ for any $i$, the MEU is not achieved.*

*Proof.* We know that $\mathbb{E}^{\boldsymbol{\pi}}[Y] = \sum_{i_{\min} \leq i \leq i_{\max}} Y^{m_i}$ (Definition 16), so for all $i$, $Y^{m_i} \leq 1$ surely. So, if $P^{\boldsymbol{\pi}}(Y^{m_i} < 1) > 0$ for any $i$, then $\mathbb{E}^{\boldsymbol{\pi}}[Y] < i_{\max} - i_{\min} + 1$, which underperforms the policy that is followed in the non-intervened model (Lemma 17). $\square$

## 4.4 Proving materiality in the materiality SCM

We will now prove that in the materiality SCM, if $Z_0$ is removed from the contexts of $X_0$, then the performance for at least one of the utility variables $Y^{m_i}$ is compromised, and so the MEU is not achieved. The proof is divided into two cases, based on whether the child of $X_0$ along the control path is a non-decision (Section 4.4.1) or a decision (Section 4.4.2).

### 4.4.1 Case 1: child of $X_0$ along $d$ is a non-decision.

We will now prove that if the child of $X_0$ along the control path is a non-decision, i.e., $X_0$ is not the context $Z_1$, then $\mathbb{E}[Y^{m_0}] < 1$. In this case, either $X_0$ is the last decision in the control path, or otherwise there must exist an intersection node $T_1$, as shown in Figure 9a. If the former is true, then it is immediate that the value $x_0$ is transmitted to $Y$ along the control path, based on the model definition. As such, $Y_0$ can directly evaluate the decision $X_0$. For the latter case, we want an assurance that downstream decisions will pass along the value of $X$, as was the case in Figure 4b. Such an assurance is provided by the following lemma, which states that whenever an intersection node $T_i$ is a chance node (as $T_1$ is) — the value $t_i$ is transmitted to $Y$ by every optimal policy.

**Lemma 19** (Chance intersection node requirement)**.** *If in the materiality SCM, where $T_i$ is a chance node, a policy $\boldsymbol{\pi}$ has $P^{\boldsymbol{\pi}}(Pa(T_i^{\boldsymbol{p}_i}) = Pa(Y^{\boldsymbol{p}_i})) < 1$, then $P^{\boldsymbol{\pi}}(Y^{m_i} < 1) > 0$.*

First, we prove the case where $m_i$ is a directed path. In this case, $m_i$ copies the value $t^{\boldsymbol{p}_i}$ to $Y$, which $Y^{m_i}$ checks against the value $\mathrm{pa}(y^{\boldsymbol{p}_i})$ received via the control path. Maximizing $Y^{m_i}$ then requires them to be equal.

*Proof of Lemma 19 when $m_i$ is a directed path.* We have $f_{Y^{m_i}}(\mathrm{pa}_{Y^{m_i}}) = [\![\mathrm{pa}(Y^{m_i}) = \mathrm{pa}(Y^{\boldsymbol{p}_i}))]\!]$ (Definition 16). Also, $\mathrm{Pa}(Y^{m_i}) = T_i^{\boldsymbol{p}_i} = \mathrm{Pa}(T_i^{\boldsymbol{p}_i})$ surely, where the first equality follows from Definition 16, while the second follows from Definition 16 and $T_i$ being a chance node. So, if $P^{\boldsymbol{\pi}}(\mathrm{Pa}(Y^{\boldsymbol{p}_i}) = \mathrm{Pa}(T^{\boldsymbol{p}_i})) < 1$, then $P^{\boldsymbol{\pi}}(Y^{m_i} = 1) < 1$. $\qquad\square$

We now prove the case where $m_i$ is a directed path. In this case, if the assignment $\mathrm{pa}(Y^{\boldsymbol{p}_i})$ transmitted along the control path differs from the value $\mathrm{pa}(T_i^{\boldsymbol{p}_i})$ that came in to the intersection node $T_i$, then just as we established for Figure 7c, there will exist an assignment $\boldsymbol{u}_{i,1:J_i}$ to the fork nodes in $m_i$ that gives an unchanged assignment to colliders $\boldsymbol{v}_{i,1:J_i}$, but where $\mathrm{pa}(Y^{\boldsymbol{p}_i})$ is incompatible with $u_{J_i}$.

*Proof of Lemma 19 when $m_i$ is not a directed path.* Let us index the forks and colliders of $m_i$ as $T_i \dashdash V_{i,1} \leftarrow\!\!\text{-} U_{i,1} \dashrightarrow W_{i,1} \leftarrow\!\!\text{-} \cdots W_{i,J_i} \leftarrow\!\!\text{-} U_{i,J_i} \dashrightarrow Y$. Choose any assignments $\mathrm{pa}(T_i^{\boldsymbol{p}_i}) \neq \mathrm{pa}(Y^{\boldsymbol{p}_i})$ that occur with strictly positive probability. Then, there must also exist assignments $\mathrm{Pa}(Y^{\boldsymbol{p}_i,\boldsymbol{r}_{i,1:J_i}}) = \mathrm{pa}(Y^{\boldsymbol{p}_i,\boldsymbol{r}_{i,1:J_i}})$, $\boldsymbol{U}_{i,1:J_i} = \boldsymbol{u}_{1:J_i}$, and $\boldsymbol{W}_{i,1:J_i} = \boldsymbol{w}_{1:J_i}$ such that

$$P^{\boldsymbol{\pi}}(\mathrm{pa}(T_i^{\boldsymbol{p}_i}), \mathrm{pa}(Y^{\boldsymbol{p}_i,r_{i,1}}), t_i^{\boldsymbol{p}_i}, \boldsymbol{u}_{1:J_i}, \boldsymbol{w}_{1:J_i}) > 0.$$

By Lemma 13, there also exists an assignment $\boldsymbol{U}_{i,1:J_i} = \boldsymbol{u}'_{1:J_i}$ such that $\mathrm{pa}(T_i^{\boldsymbol{p}_i}), \boldsymbol{w}_{1:J_i}$ is consistent with $\boldsymbol{u}'_{1:J_i}$, and $\mathrm{pa}(Y_i^{\boldsymbol{p}}), \mathrm{pa}(Y^{\boldsymbol{r}_{i,1:J_i}})$ is incompatible with $\boldsymbol{u}'_{J_i}$. Now, consider the intervention $\mathrm{do}(\boldsymbol{U}_{i,1:J_i} = \boldsymbol{u}'_{1:J_i})$. Since $T_i$ is a chance node, every collider in $m_i$ is a non-decision, and is assigned the (unique) value consistent with $\mathrm{pa}(T_i^{\boldsymbol{p}_i}), \boldsymbol{u}'_{1:J_i}$. Furthermore, $\mathrm{pa}(T_i^{\boldsymbol{p}_i}), \boldsymbol{w}_{1:J_i}$ is consistent with $\mathrm{pa}(T_i^{\boldsymbol{p}_i}), \boldsymbol{u}'_{1:J_i}$, so the intervention does not affect the assignments to these colliders. Moreover, from Definition 16, no variable outside of $m_i$ is affected by assignments within $m_i$, except through the colliders. Therefore:

$$P^{\boldsymbol{\pi}}(\mathrm{pa}(Y^{\boldsymbol{p}_i}), \mathrm{pa}(Y^{\boldsymbol{r}_{i,1:J_i}}), \mathrm{Pa}(Y^{m_i}) = \boldsymbol{u}'_{J_i} \mid \mathrm{do}(\boldsymbol{U}_{i,1:J_i} = \boldsymbol{u}'_{1:J_i})) > 0$$

$$\therefore P^{\boldsymbol{\pi}}(Y^{m_i} = 0 \mid \mathrm{do}(\boldsymbol{U}_{i,1:J_i} = \boldsymbol{u}'_{1:J_i})) > 0$$

$$(\mathrm{pa}(Y_i^{\boldsymbol{p}}), \mathrm{pa}(Y^{\boldsymbol{r}_{i,1:J_i}}) \text{ not compatible with } \boldsymbol{u}'_{J_i})$$

$$\therefore P^{\boldsymbol{\pi}}(Y^{m_i} = 0 \mid \boldsymbol{U}_{i,1:J_i} = \boldsymbol{u}'_{1:J_i}) > 0$$

$$(\boldsymbol{U}_{i,1:J_i} \text{ are unconfounded, so } P^{\boldsymbol{\pi}}(\boldsymbol{V} \mid \mathrm{do}(\boldsymbol{U}_{i,1:J_i} = \boldsymbol{u}'_{1:J_i})) = P^{\boldsymbol{\pi}}(\boldsymbol{V} \mid \boldsymbol{U}_{i,1:J_i} = \boldsymbol{u}'_{1:J_i}))$$

$$\therefore P^{\boldsymbol{\pi}}(Y^{m_i} = 0) > 0 \qquad\qquad (P^{\boldsymbol{\pi}}(\boldsymbol{u}'_{i,1:J_i}) > 0).$$

$\qquad\square$

If $m_i$ is not a directed path, then we will require that the values $\mathrm{pa}(Y^{\boldsymbol{r}_{i,1:J_i}})$ are passed down the auxiliary paths, not just the value $\mathrm{pa}(Y^{\boldsymbol{p}_i})$ from the control path. Specifically, $\mathrm{pa}(Y^{\boldsymbol{p}_i}), \mathrm{pa}(Y^{\boldsymbol{r}_{i,1:J_i}})$ must be consistent with $\mathrm{pa}(Y^{\boldsymbol{p}_i}), \boldsymbol{u}_{i,1:J_i}$, where $\boldsymbol{u}_{i,1:J_i}$ denotes the values of forks on the info path.

**Lemma 20** (Collider path requirement). *If the materiality SCM has an info path $m_i$ that is not directed, and under the policy $\boldsymbol{\pi}$ there are assignments $Pa(Y^{\boldsymbol{p}_i, \boldsymbol{r}_{i,1:J_i}}) = pa(Y^{\boldsymbol{p}_i, \boldsymbol{r}_{i,1:J_i}})$ to parents of the outcome, and $\boldsymbol{U}_{i,1:J_i}^{m_i} = \boldsymbol{u}_{i,1:J_i}^{m_i}$ to the forks of $m_i$, with $P^{\boldsymbol{\pi}}(pa(Y^{\boldsymbol{p}_i, \boldsymbol{r}_{i,1:J_i}}), \boldsymbol{u}_{i,1:J_i}^{m_i}) > 0$ and where $pa(Y^{\boldsymbol{p}_i, \boldsymbol{r}_{i,1:J_i}})$ is inconsistent with $pa(Y^{\boldsymbol{p}_i}), \boldsymbol{u}_{i,1:J_i}^{m_i}$, then $P^{\boldsymbol{\pi}}(Y^{m_i} < 1) > 0$.*

The idea of the proof, similar to Lemma 19, is that whenever the bits transmitted along the auxiliary paths deviate from the values $\boldsymbol{w}_{i,1:J_i}$ of colliders in $m_i$, there exists an assignment $\boldsymbol{u}'_{i,1:J_i}$ to forks in $m_i$ that will render the colliders, and hence the decision $x_i$ unchanged, while making $x_i$ incompatible with $u_{J_i}$, and thereby producing $Y^{m_i} < 0$. A detailed proof is in Section B.5.

Before we prove that $Z_0$ is material in this case, we require one more intermediate result: that $Z_0$ cannot be chosen deterministically, if it is a decision. The idea will be that random information is generated at $A$, which each of the decisions, including $Z_0$, is required to pass along the control path; we can prove this as a corollary of Lemma 19.

**Lemma 21** (Initial truncated info path requirements). *If $\boldsymbol{\pi}$ in the materiality SCM does not satisfy: $P^{\boldsymbol{\pi}}(Pa(Y^d) = A^d) < 1$. then the MEU is not achieved.*

*Proof.* From Lemma 11, the control path $d$ begins with a chance node. So, the first decision $X_{i_{\min}}$ in $d$ must have a chance node $Z_{i_{\min}}$ as its parent along $d$. Furthermore, the intersection node $T_{i_{\min}}$ must be an ancestor of $Z_{i_{\min}}$ along $d$, so it is also a chance node. So it follows from Lemma 19, that any policy $\boldsymbol{\pi}$ must satisfy $P^{\boldsymbol{\pi}}(T_{i_{\min}}^{\boldsymbol{p}_{i_{\min}}} = \mathrm{Pa}(Y^{\boldsymbol{p}_{i_{\min}}})) = 1$ if it attains the MEU. As $T_{i_{\min}}$ is in the control path, we have $d \in \boldsymbol{p}_{i_{\min}}$ (by Lemma 11) so $T_{i_{\min}}^d \overset{\text{a.s.}}{=} \mathrm{Pa}(Y^d)$ is also required. Moreover, all of vertices in the segment $A \dashrightarrow T_{i_{\min}}$ of $d$ are chance nodes, because $X_{i_{\min}}$ was defined as the first decision in $d$, and $T_{i_{\min}}$ precedes it. And, each chance variable $V^d$ on the control path equals its parent $\mathrm{Pa}(V^d)$ (by Definition 16), so $A^d = T_{i_{\min}}^d$, and thus $A^d \overset{\text{a.s.}}{=} \mathrm{Pa}(Y^d)$ is required to attain the MEU. $\qquad\square$

We can now combine our previous results to prove that it is impossible to achieve the MEU, if $Z_0$ is not a context of $X_0$, in the case where $T_1$ does not exist, or is a non-decision.

**Lemma 22** (Required properties unachievable if child is a non-decision). *Let $\mathcal{M}$ be a materiality SCM where the child of $X_0$ along $d$ is a non-decision. Then, the MEU for the scope $\mathcal{S}$ cannot be achieved by a deterministic policy in the scope $\mathcal{S}_{Z_0 \not\rightarrow X_0}$ (equal to $\mathcal{S}$, except that $Z_0$ is removed from $\boldsymbol{C}_{X_0}$).*

The logic is that if child of $X_0$ in the control path is a non-decision, then the value of $X_0$ is copied all the way to $\mathrm{Pa}(Y^d)$ (Lemma 21). Furthermore, $Z_0^d \overset{\text{a.s.}}{=} \mathrm{Pa}(Y^d)$ is necessary to achieve the MEU (Lemma 19). But the materiality SCM has been constructed so that the non-$Z_0$ parents of $X_0$ do not contain enough bits to transmit all of the information about $Z_0^d$, so the MEU cannot be achieved. The proof is detailed in Section B.6.

### 4.4.2 Case 2: child of $X_0$ along $d$ is a decision.

We will now prove that if, as shown in Figure 9b, the child of $X_0$ along $d$ is a decision and the decision $Z_0$ is not available as a context of $X_0$, then $\mathbb{E}[Y_1] < 1$. The overall argument will be that if $X_0$ cannot observe $Z_0$, then it cannot convey the value of $U_{1,1}$ to the child $X_1$, and so $X_1$ will not be able to be consistent with $\boldsymbol{u}_{0:J_1}$.

Our first lemma will state that if a decision $X_{i-1}$ does not distinguish $U_{i,1}$, and $U_{i,2}$ is a bitstring wherein all of the digits are the same, then $U_{i,1}$ cannot influence $Y$, other contexts of $T_i$, colliders $\boldsymbol{W}_{i,1:J_i}$, or forks $U_{i,2:J_i}$. A couple of details of this lemma are worth remarking upon. First, it assumes that $T_i = X_{i-1}$ — something that is always true for $i = 1$ in the case that $Z_1 = X_0$. Second, it assumes that $\pi_{T_i}$ is a deterministic decision rule, which will simplify the proofs without importantly weakening it, because every decision problem has a deterministic policy that performs as well as the best non-deterministic policy.

**Lemma 23** (If next fork is repeated, then fork only influences intersection node). *If, in the materiality SCM:*

- *the intersection node $T_i$ is the vertex $X_{i-1}$,*

- $\pi_{T_i}$ *is a deterministic decision rule where* $\pi_{T_i}(\boldsymbol{c}^{\neg m_i}(T_i, u_{i,1}) = \pi_{T_i}(\boldsymbol{c}^{\neg m_i}(T_i, u'_{i,1}))$ *for assignments* $u_{i,1}, u'_{i,1}$ *to the first fork variable, and* $\boldsymbol{c}^{\neg m_i}(T_i)$ *to the contexts of* $T_i$ *not on* $m_i$, *and*
- $\boldsymbol{W}_{i,1:J_i} = \boldsymbol{w}_{i,1:J_i}$, *and* $\boldsymbol{U}_{i,2:J_i} = \boldsymbol{u}_{i,2:J_i}$ *are assignments to forks and colliders in* $m_i$ *where each* $u_{i,j}$ *consists of just* $w_{i,j}$ *repeated* $\exp_2^j(k + |\boldsymbol{p}_i| - 1)$ *times, then:*

$$P^{\boldsymbol{\pi}}(pa(Y^{\boldsymbol{p}_i, r_{i,1}}), \boldsymbol{c}^{\neg m_i}(T_i), \boldsymbol{w}_{i,1:J_i}, \boldsymbol{u}_{i,2:J_i} \mid do(u_{i,1})) = P^{\boldsymbol{\pi}}(pa(Y^{\boldsymbol{p}_i, r_{i,1}}), \boldsymbol{c}^{\neg m_i}(T_i), \boldsymbol{w}_{i,1:J_i}, \boldsymbol{u}_{i,2:J_i} \mid do(u'_{i,1})).$$

The proof follows from the definition of the materiality SCM, and it is detailed in Section B.7.

We can now prove that if a deterministic policy does not appropriately distinguish assignments to $U_{i,1}$, then the $i^{\text{th}}$ component of the utility will be suboptimal, i.e., $\mathbb{E}[Y^{m_i}] < 1$.

**Lemma 24** (Decision must distinguish fork values)**.** *If in the materiality SCM:*

- *the intersection node* $T_i$ *is the vertex* $X_{i-1}$, *and*
- $\boldsymbol{\pi}$ *is a deterministic policy that for assignments* $u_{i,1}, u'_{i,1}$ *to* $U_{i,1}$ *where* $u_{i,1} \neq u'_{i,1}$,        (†)
  *has* $\pi_{T_i}(\boldsymbol{c}^{\neg m_i}(T_i), u_{i,1}) = \pi_{T_i}(\boldsymbol{c}^{\neg m_i}(T_i), u'_{i,1})$ *for every* $\boldsymbol{C}_{T_i}^{\neg m_i}(T_i) = \boldsymbol{c}^{\neg m_i}(T_i)$,

*then* $P^{\boldsymbol{\pi}}(Y^{m_i} < 1) > 0$

The idea of the proof is that if $u_{i,1}$ and $u'_{i,1}$ differ, there will be some assignment $pa(Y^{\boldsymbol{p}_i})$ such that $u_{i,1}[pa(Y^{\boldsymbol{p}_i})]$ and $u'_{i,1}[pa(Y^{\boldsymbol{p}_i})]$ differ. When $\text{Pa}(Y^{\boldsymbol{p}_i}) = pa(Y^{\boldsymbol{p}_i})$ and $U_{i,1} = u_{i,1}$, then $\text{Pa}(Y^{r_{i,1}})$ will assume one value. But if we intervene $u'_{i,1}, u_{i,2:J_i}$, then the value of $\text{Pa}(Y^{r_{i,1}})$ will be incorrect, making $\text{Pa}(Y^{\boldsymbol{p}_i, \boldsymbol{r}_{i,1:J_i}})$ inconsistent with $\text{Pa}(Y^{\boldsymbol{p}_i}, U_{i,1:J_i})$ so the maximum expected utility is impossible to achieve. The details are deferred to Section B.8.

This allows us to prove that when the child of $X_0$ along $d$ is a decision, the MEU cannot be achieved unless $Z_0$ is a context of $X_0$.

**Lemma 25** (Required properties unachievable if child is a decision)**.** *Let* $\mathcal{M}$ *be the materiality SCM for some scoped graph* $\mathcal{G}_{\mathcal{S}}$, *where* $i_{max} > 0$ *and* $T_1$ *is a decision. Then, there exists no deterministic policy in the scope* $\mathcal{S}_{Z_0 \not\rightarrow X_0}$ *that achieves the MEU.*

To prove that no deterministic policy in $\mathcal{S}_{Z_0 \not\rightarrow X_0}$ can achieve the MEU (achievable with the scope $\mathcal{S}$), we will show that if a deterministic policy $\boldsymbol{\pi}$ satisfies $P^{\boldsymbol{\pi}}(\text{Pa}(Y^d) = A^d) = 1$, as required by Lemma 21, then the domain of $X_0 \times C_{X_0}^{\neg m_1}$ is smaller than the domain of $C_{X_0}^{m_1}$, so Equation (†) will be satisfied, and thus the MEU cannot be achieved. A detailed proof is presented in Section C.

We now combine Lemmas 22 and 25 to prove the main result.

*Proof of Theorem 8.* Any scoped graph $\mathcal{G}(\mathcal{S})$ that satisfies conditions (A-C) contains materiality paths for the context $Z_0$ of $X_0$ (Lemma 11), and has a materiality SCM (Definition 16) compatible with $\mathcal{G}(\mathcal{S})$. In this decision problem, whether the child of $X_0$ along $d$ is or is not a decision, the MEU cannot be achieved by a deterministic policy unless $X_0$ is allowed to depend on $Z_0$ (Lemmas 22 and 25). And stochastic policies can never surpass the best deterministic policy (Lee & Bareinboim, 2020, Proposition 1), so no such policy can achieve the MEU, and therefore $Z_0$ is material for $X_0$. $\qquad\square$

## 5   Toward a more general proof of materiality

So far, via Theorem 8, we have established the necessity of condition (I) of LB-factorizability for immateriality. We now outline some steps toward evaluating the necessity of conditions (II-III) of LB-factorizability, as well as the further condition in (Lee & Bareinboim, 2020, Thm. 2).

It is trivial to satisfy either one of (II-III) in isolation. Condition (III) merely requires that we choose an ordering $\prec$ such that the parents of each decision $X$ are prior to $X$, while the descendants come afterwards, and such an ordering clearly exists in any acyclic graph. Condition (II) can also be satisfied by placing all of the variables in $\boldsymbol{C}$ at the start of the ordering $\prec$.

However, there does not always exist an ordering that satisfies (II-III) simultaneously. Indeed, whenever there does not, we will be able to prove the existence of some info paths and control paths. If we could use these paths to establish materiality, then we would have proved that (II-III) are necessary conditions. So far, however, we have only been able to carry out the first step — defining the paths — and difficulties have arisen in using those paths to define an SCM that exhibits materiality. In this section, we will outline what info paths and control paths can be proven to exist, and then outline the difficulties in using them to prove materiality.

### 5.1 A lemma for proving the existence of paths

When the variables $\boldsymbol{Z}, \boldsymbol{X'}, \boldsymbol{C'}, \boldsymbol{U}$ are not factorizable, we can prove the existence of info and control paths.

**Lemma 26** (System Exists General). *Let $\mathcal{G}_\mathcal{S}$ be a scoped graph that satisfies conditions (A,B) from Theorem 8. If $\boldsymbol{Z} = \{Z\}$, $\boldsymbol{X'} \supseteq Ch(Z)$, $\boldsymbol{C'} = C_{\boldsymbol{X'}} \setminus (\boldsymbol{X'} \cup \boldsymbol{Z})$, $\boldsymbol{U} = \emptyset$ are not LB-factorizable, then there exists a pair of paths to some $C' \in \boldsymbol{C'} \cup Y$:*

- *an info path $m : Z \dashrightarrow C'$, active given $\lceil \boldsymbol{X'} \cup \boldsymbol{C'} \rceil$, and*
- *a control path $d : X \dashrightarrow C'$ where $X \in \boldsymbol{X'}$.*

A proof is supplied in Section D.1. The intuition of this proof is that each of the conditions (I-III) implies a precedence relation between a pair of variables in $\boldsymbol{V'} \cup Y$. Each of these precedence relations can be used to build an "ordering graph" over $\boldsymbol{V'} \cup Y$. If the ordering graph is acyclic, then we can let $\prec$ be any ordering that is topological on the graph, and then $\boldsymbol{Z}, \boldsymbol{X'}, \boldsymbol{C'}, \boldsymbol{U}$ are LB-factorizable. Otherwise, we can use a cycle in the graph to prove the existence of an info path and a control path. By iterating over these cycles, we can obtain a series of info paths and control paths that terminate at $Y$.

The resulting paths are in some cases quite useful for proving materiality. For instance, we can recover the pair of info and control paths used in Figure 4b. To prove that $Z$ is material for $X$, we can start by choosing $\boldsymbol{X'} = \{X, X'\}$, $\boldsymbol{C'} = \{Z', W\}$, and $\boldsymbol{U'} = \emptyset$. Then, Lemma 26 implies the existence of an active path from $Z$ to some $\text{Desc}_X \cap \boldsymbol{C'}$, so we see that the first info path is the edge $Z \rightarrow Y$. Since $Y$ is a descendant of $X$, we also have the first control path, $X \rightarrow Z' \rightarrow X' \rightarrow Y$. We must then obtain some paths that exhibit why $Z'$ is itself useful for the decision $X$ to know about, and to influence. To do this, we can reapply Lemma 26 using the sets $\boldsymbol{X'} = \{X, X'\}, \boldsymbol{Z} = \{Z'\}, \boldsymbol{C'} = \{Z, W\}$, and $\boldsymbol{U'} = \emptyset$. We then obtain the new info path $Z' \rightarrow W \leftarrow U \rightarrow Y$, and the new control path $X' \rightarrow Y$. The SCM in Figure 4b uses these paths to prove $Z$ is material for $X$.

### 5.2 A further challenge: non-collider contexts

In some graphs, it is not clear how to use the info and control paths from Lemma 26 to prove materiality, because non-collider nodes on the info path may be contexts. (In previous work, this possibility was excluded by the solubility assumption (van Merwijk et al., 2022, Lemma 28).) We will now highlight one case, in Figure 10, where it is relatively clear how this challenge can be overcome, and one case, Figure 11, where it is unclear how to make progress.

In the graph of Figure 10, we would like to prove that $Z_0$ is material for $X_0$. Using Lemma 26, we can obtain the red and blue info paths as shown, and the corresponding control paths, shown in darker versions of the same colours. The usual idea of Definition 16, shown in Figure 10a, would be that $X_0$ must observe $Z_0$ in order to know which slice from $V$ is presented at its parent $X_1$. In this model, $X_1$ would play two roles, one for the red info path, and one for the dark blue control path. As a collider on the red info path, its role is to present the $Z_0^{\text{th}}$ bit from $V$. As the initial endpoint of the blue control path, its role is to copy the assignment of $Z_0$. The problem, however, is that $X_0$ then does not need to observe $Z_0$ in order to reproduce its value, because this value is already observed at $X_1$, and so $Z_0$ is not material.

To remedy this problem, we can construct an alternative SCM, where the value of $Z_0$ is "concealed", i.e. it is removed from the other contexts, $C_{Z_0} \setminus Z_0$. We now define $X_1$ as $V[Z_0]$ (i.e. we have removed $Z_0$), leaving the decision $X_1$ with a domain of only one bit. At $C$, we impose some random noise; that is $C$ is

not always a perfect copy of $Z_0$. The result is shown in Figure 10b. When this model is not intervened, an expected utility of $\mathbb{E}[Y] = 10.99$ is achieved, because the red term in $Y$ always equals 10, while the blue term has an expectation of 0.99. (This is the MEU, because there is no way to improve the blue term to have expectation 1 without decreasing the expectation of the red term by at least 0.05.) If instead $Z_0$ is removed as a context for $X_0$, then the expected utility can only be as high as $\mathbb{E}[Y] = 10.95$. To understand this, we restrict our attention to deterministic policies, and note that in order for the red term to be better than a coin flip (with an expected value of 5), we would either need to have $X_0 = \langle C, X_1 \rangle$ — and the red term will have an expectation of 9.95, or we must have $X_1 = V[0]$ and $X_0 = \langle 0, X_1 \rangle$ — and then the blue term will have an expectation of 0.5. In either case, performance is worse than 10.99, so $Z_0$ is material for $X_0$.

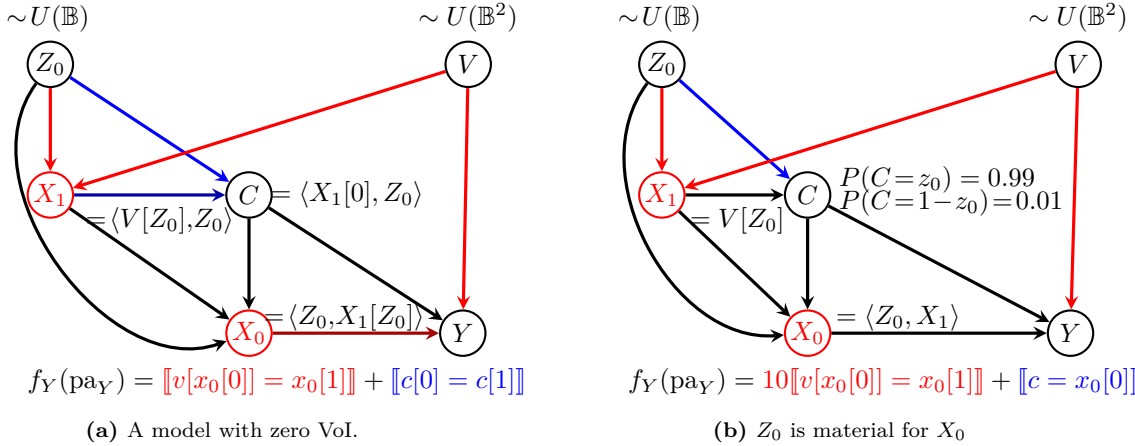

**(a)** A model with zero VoI.  **(b)** $Z_0$ is material for $X_0$

**Figure 10:** Two alternative models that use the same two info paths, red and blue.

The problem is that concealing the value of $Z_0$ does not work for all graphs. To see this, let us add two decisions, $X_2$ and $X_3$, to the graph from Figure 10, to thereby obtain the graph in Figure 11. Let us retain the materiality SCM from Figure 10b, except that $X_2$ and $X_3$ copy the value from $C$ along to $Y$. One might expect that $Z_0$ should still be material, but it is not. Now, there is a policy that achieves the new MEU of 11 by superimposing the value of $Z_0$ on the assignments of decisions $X_2$ and $X_3$. In this policy $\boldsymbol{\pi}$, $x_1 = v[z_0]$, $x_2 = z_0 \oplus z_0$, $x_3 = x_2 \oplus z_0$, and $x_0 = x_2 \oplus x_3 = z_0$ where $\oplus$ represents the XOR function. Under $\boldsymbol{\pi}$, the red term equals 10 surely, while the blue term equals 1 surely, i.e. the MEU is achieved, and $\boldsymbol{\pi}$ is a valid policy even if $Z_0$ is not a context of $X_0$, meaning that $Z_0$ is not material for $X_0$.

In summary, whenever $\boldsymbol{Z} \ni Z_0, \boldsymbol{X'} \ni X_0, \boldsymbol{C'}, \boldsymbol{U}$ are not LB-factorizable, then we can find some info and control paths for $Z_0$ and $X_0$, but then $X_0$ can recover the value of $Z_0$, making it possible to achieve the MEU even when $Z_0$ is removed as a context of $X_0$. In some graphs, we can devise an alternative SCM that conceals the value of $Z_0$. But in others, a policy can superimpose the information from $Z_0$ on other decisions, such as $X_2$ and $X_3$ in Figure 11, so that $X_0$ can recover the value of $Z_0$, making $Z_0$ immaterial for $X_0$ once again.

Overall, in order to establish a complete criterion for materiality, we would need some new method to prevent the information from $Z_0$ from being superimposed on other decisions. So, in order for future work to achieve this goal, we predict that it will have to make further modifications to the construction from Definition 16.

## 6 Conclusion

In graphical models of decision-making, a key challenge is to ascertain which variables can be deemed immaterial based on the graphical structure alone. This problem is a long-standing one, a solution to which could allow influence diagrams to be solved more efficiently, and aid in analyzing the safety and fairness of AI systems. A key condition for establishing immateriality is LB-factorizability. We have found that if no context can satisfy condition (I) of LB-factorizability, then any context is material in at least one decision

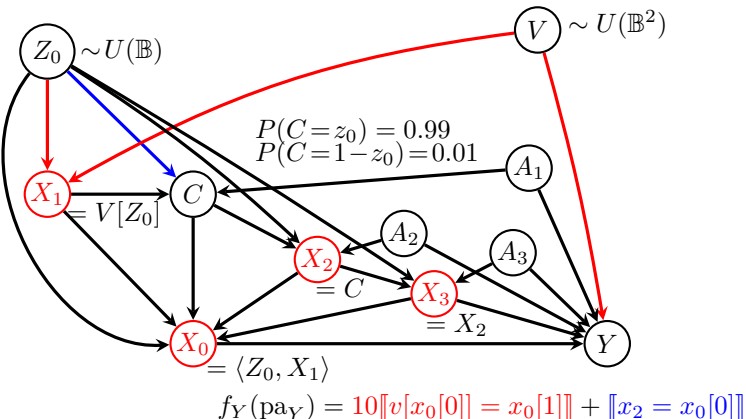

**Figure 11:** A model with zero VoI

problem that respects the graphical constraints. In the process of devising our proofs of materiality, we encountered and addressed several new kinds of problems:

- if, as in Figure 5, the variable $Z_i$ whose materiality we are trying to establish is a decision, then we must choose an info path so that the value of $Z_i$ cannot be determined by other contexts;
- if, as in Figure 7, the info path begins with a variable $Z_0$ that is a context not just of the target decision $X_0$, but also some other decision $X'$, then we must construct the decision problem so that $X'$ cannot copy $Z_0$ in its entirety;
- if, as in Section 4.4.2, the control path contains consecutive decisions, then we give large domains to the source $A$, and all of the variables on the control path, so that this information can only possibly be copied by these decisions.

As a next step towards establishing a complete criterion for materiality, we then considered the more general setting where no context can jointly satisfy conditions (I-III) of LB-factorizability. In this setting, we have identified info paths and control paths that appear useful for proofs of materiality, and we can apply an adjusted version of our construction to this case. However, in some settings there are ways for optimal performance to be achieved even when $Z_0$ is not a context of $X_0$, and so we have not yet found a successful proof of materiality in this more general setting.

Thus, the challenge of proving a complete criterion for materiality for insoluble graphs currently remains open.

## Acknowledgements

Thanks to Min Woo Park and Tom Everitt for comments on draft versions of this manuscript. Sang-hack Lee was supported by the IITP (RS-2022-II220953/25%) and NRF (RS-2023-00211904/50%, RS-2023-00222663/25%) grant funded by the Korean government.

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
