# A    Recap of Lee and Bareinboim (2020)

Our result Theorem 8 is an initial step in a larger potential project of proving that (Lee & Bareinboim, 2020, Thm. 2) is a complete criterion for materiality.

The result begins with the following factorization, of which we are only focused on cases where the first condition is violated (Lee & Bareinboim, 2020). The result uses the definition of "redundancy" (which is a looser condition than immateriality): if a scoped graph $\mathcal{G}(\mathcal{S})$ has $X \not\subseteq \mathrm{Anc}_Y$ or $(C \not\perp Y \mid X \cup \mathrm{Pa}_X \setminus \{C\})$ then it is "redundant", The result from Lee & Bareinboim (2020) is reproduced verbatim:

*Lemma* LB-1. Given an MPS $\mathcal{S}$, which satisfies non-redundancy, let $\boldsymbol{X}' \subseteq \boldsymbol{X}(\mathcal{S})$, actions of interest, $\boldsymbol{C}' \subsetneq \boldsymbol{C}_{\boldsymbol{X}'} \setminus \boldsymbol{X}'$. non-action contexts of interest. If there exists a subset of exogenous variables $\boldsymbol{U}'$ in $\mathcal{G}_{\mathcal{S}}$, a subset of endogenous variables $\boldsymbol{Z}$ in $\mathcal{G}_{\mathcal{S}}$ that is disjoint with $\boldsymbol{C}' \dot{\cup} \boldsymbol{X}'$ and subsumes $\boldsymbol{C}_{\boldsymbol{X}'} \setminus (\boldsymbol{C}' \dot{\cup} \boldsymbol{X}')$, and an order $\prec$ over $\boldsymbol{V}' \doteq \boldsymbol{C}' \dot{\cup} \boldsymbol{X}' \dot{\cup} \boldsymbol{Z}$ such that

1. $(Y \perp \boldsymbol{\pi}_{\boldsymbol{X}'} \mid \lceil \boldsymbol{X}' \dot{\cup} \boldsymbol{C}' \rceil)_{\mathcal{G}_{\mathcal{S}}}$,
2. $(C \perp \boldsymbol{\pi}_{\boldsymbol{X}'_{\prec C}}, \boldsymbol{Z}_{\prec C}, \boldsymbol{U}' \mid \lceil (\boldsymbol{X}' \dot{\cup} \boldsymbol{C}')_{\prec C} \rceil)_{\mathcal{G}_{\mathcal{S}}}$ for every $C \in \boldsymbol{C}'$, and
3. $\boldsymbol{V}'_{\prec X}$ is disjoint with $de(X)_{\mathcal{G}_{\mathcal{S}}}$ and subsumes $pa(X)_{\mathcal{G}_{\mathcal{S}}}$ for every $X \in \boldsymbol{X}'$,

where, the policy node $\pi_X$ is a new parent added to $X$, then the expected reward for $\boldsymbol{\pi}$, a deterministic policy optimal with respect to $\mathcal{S}$, can be written as

$$\mu_{\boldsymbol{\pi}} = \sum_{y,\boldsymbol{c}',\boldsymbol{x}'} y Q'_{\boldsymbol{x}'}(y, \boldsymbol{c}') \sum_{\boldsymbol{u}',\boldsymbol{z}} Q(\boldsymbol{u}') \prod_{Z \in \boldsymbol{Z}} Q(z | \boldsymbol{v}'_{\prec z}, \boldsymbol{u}') \prod_{X \in \boldsymbol{X}'} \pi(x | \boldsymbol{c}_x). \tag{1}$$

Lemma LB-1 provides conditions for asserting Equation (1) given $(\mathcal{S}, \boldsymbol{X}', \boldsymbol{C}')$, whether $(\boldsymbol{U}', \boldsymbol{Z}, \prec)$ exist satisfying three conditions. It is then used to prove redundancy under optimality using the following theorem.

*Theorem* LB-2. Let $\boldsymbol{U}', \boldsymbol{Z}$ and $\prec$ satisfy Lemma LB-1. For $Z \in \boldsymbol{Z}$, let $\boldsymbol{V}_Z$ be a minimal subset of $\boldsymbol{V}'_{\prec Z} \cup \boldsymbol{U}'$ such that $Z \perp \boldsymbol{U}' \mid \boldsymbol{V}_Z$. We define fix($\boldsymbol{T}$) with respect to $\{\langle Z, \boldsymbol{V}_Z \rangle\}_{Z \in \boldsymbol{Z}}$, that is with $\hat{\boldsymbol{T}} := \lceil \boldsymbol{T} \rceil \cup \{Z \in \boldsymbol{Z} \mid \boldsymbol{V}_Z \setminus \boldsymbol{U}' \subseteq \lceil \boldsymbol{T} \rceil \}$, and fix($\boldsymbol{T}$) is $\boldsymbol{T}$ if $\boldsymbol{T} = \hat{\boldsymbol{T}}$, and fix($\hat{\boldsymbol{T}}$) otherwise. If fix($\boldsymbol{C}_X \setminus \boldsymbol{Z}$) $\supseteq \boldsymbol{C}_X$ for $X \in \boldsymbol{X}'$, then $\mathcal{S}' := (\mathcal{S} \setminus \boldsymbol{X}') \cup \{\langle X, \boldsymbol{C}_X \setminus \boldsymbol{Z} \rangle\}_{X \in \boldsymbol{X}'}$ satisfies $\mu^*_{\mathcal{S}'} = \mu^*_{\mathcal{S}}$.

Let us apply Theorem LB-2 to the graph Figure 2, which we discussed in Section 2. We have noted that using $\boldsymbol{Z} = \{Z\}, \boldsymbol{X}' = \{X\}$, and the ordering $\prec = \langle Z, X \rangle$, $\boldsymbol{Z}$ and $\boldsymbol{X}'$ are LB-factorizable. To apply the theorem, we must confirm that fix($\boldsymbol{C}_X \setminus \boldsymbol{Z}$) $\supseteq \boldsymbol{C}_X$ is true. The right hand side is simply equal to $Z$. To evaluate the left hand side, note that $\boldsymbol{C}_X \setminus \boldsymbol{Z} = \emptyset$. Furthermore, $\hat{\emptyset}$ includes $\lceil \boldsymbol{T} \rceil$, which includes $Z$. So fix($\emptyset$) also includes $Z$, meaning that the left hand side, fix($\emptyset$) is a superset of the right hand side, $\boldsymbol{C}_X$, and thus $Z$ is immaterial for $X$.

An interested reader may refer to Lee & Bareinboim (2020) for further examples where LB-2 is used to establish immateriality.

# B    Supplementary proofs regarding the main result (Theorem 4)

## B.1    Proof of Lemma 10

We begin by restating the lemma.

**Lemma 10.** *If a scoped graph $\mathcal{G}(\mathcal{S})$ satisfies conditions (B-C) of Theorem 8, then for every edge $Z \to X$ between decisions $Z, X \in \boldsymbol{X}(\mathcal{S})$, there exists a path $Z \leftarrow N \dashrightarrow Y$, active given $\lceil (\boldsymbol{X}(\mathcal{S}) \cup C_{\boldsymbol{X}(\mathcal{S}) \setminus \{Z\}}) \setminus \{Z\} \rceil$, (so $N \notin \lceil (\boldsymbol{X}(\mathcal{S}) \cup C_{\boldsymbol{X}(\mathcal{S}) \setminus \{Z\}}) \setminus \{Z\} \rceil$).*

We now prove Lemma 10.

*Proof of Lemma 10.* Since $Z$ is assumed to be a decision, we have from Lemma 9, that there exists $N \in \mathrm{Pa}_Z \setminus \lceil (\boldsymbol{X}(\mathcal{S}) \cup C_{\boldsymbol{X}(\mathcal{S}) \setminus Z}) \setminus Z \rceil$, which therefore is also a chance node. Condition (C) of Theorem 8 for

$N \to Z$ implies the existence of a path $p : \Pi_Z \to Z \leftarrow N \dashrightarrow Y$ active given $\lceil (\boldsymbol{X}(\mathcal{S}) \cup C_{\boldsymbol{X}(\mathcal{S}) \setminus N}) \setminus N \rceil$, which can be truncated as $p' : Z \leftarrow N \overset{p}{\dashrightarrow} Y$. We will consider the cases where every collider in $p'$ is in $\lceil (\boldsymbol{X}(\mathcal{S}) \cup C_{\boldsymbol{X}(\mathcal{S}) \setminus Z}) \setminus Z \rceil$, or there exists one that is not.

Case 1. Every collider in $p'$ is in $\lceil (\boldsymbol{X}(\mathcal{S}) \cup C_{\boldsymbol{X}(\mathcal{S}) \setminus Z}) \setminus Z \rceil$. Clearly $p'$ begins as $Z \leftarrow \cdot$ and terminates at $Y$ and is active at colliders, given $\lceil (\boldsymbol{X}(\mathcal{S}) \cup C_{\boldsymbol{X}(\mathcal{S}) \setminus Z}) \setminus Z \rceil$. We will now prove that $p'$ is also active given $\lceil (\boldsymbol{X}(\mathcal{S}) \cup C_{\boldsymbol{X}(\mathcal{S}) \setminus Z}) \setminus Z \rceil$ at non-colliders. Note that $\lceil (\boldsymbol{X}(\mathcal{S}) \cup C_{\boldsymbol{X}(\mathcal{S}) \setminus N}) \setminus N \rceil = \lceil (\boldsymbol{X}(\mathcal{S}) \cup C_{\boldsymbol{X}(\mathcal{S})}) \setminus N \rceil \supseteq \lceil (\boldsymbol{X}(\mathcal{S}) \cup C_{\boldsymbol{X}(\mathcal{S}) \setminus Z}) \setminus (Z \cup N) \rceil = \lceil (\boldsymbol{X}(\mathcal{S}) \cup C_{\boldsymbol{X}(\mathcal{S}) \setminus Z}) \setminus Z \rceil$, where the first equality follows from $N$ being a chance node, and the latter follows from that and $N \notin \boldsymbol{C}_{\boldsymbol{X}(\mathcal{S}) \setminus Z}$, which jointly imply that $N \in \mathrm{Pa}_Z \setminus \lceil (\boldsymbol{X}(\mathcal{S}) \cup C_{\boldsymbol{X}(\mathcal{S}) \setminus Z}) \setminus Z \rceil$. So $p'$ is active given $\lceil (\boldsymbol{X}(\mathcal{S}) \cup C_{\boldsymbol{X}(\mathcal{S}) \setminus Z}) \setminus Z \rceil$ at non-colliders, and the result is proved for this case.

Case 2. There exists a collider in $p'$ that is not in $\lceil (\boldsymbol{X}(\mathcal{S}) \cup C_{\boldsymbol{X}(\mathcal{S}) \setminus Z}) \setminus Z \rceil$. Let $M$ be the collider in $p'$ that is not in $\lceil (\boldsymbol{X}(\mathcal{S}) \cup C_{\boldsymbol{X}(\mathcal{S}) \setminus Z}) \setminus Z \rceil$, nearest to $Z$ along $p'$. Since $p'$ is active given $\lceil (\boldsymbol{X}(\mathcal{S}) \cup C_{\boldsymbol{X}(\mathcal{S}) \setminus N}) \setminus N \rceil$, we have $M \in \mathrm{Anc}_{\lceil (\boldsymbol{X}(\mathcal{S}) \cup C_{\boldsymbol{X}(\mathcal{S}) \setminus N}) \setminus N \rceil}$, which implies $M \in \boldsymbol{X}(\mathcal{S}) \cup (C_{\boldsymbol{X}(\mathcal{S})}$ (because $M \in \lceil \boldsymbol{W} \rceil \setminus \boldsymbol{W} \implies M \in \boldsymbol{X}(\mathcal{S}))$, so $M$ is an ancestor of some decision $X'$. By condition (B) of Theorem 8, $X'$ is an ancestor of $Y$, so we can construct $p'' : Z \overset{p'}{\dashrightarrow} M \dashrightarrow X' \dashrightarrow Y$, and prove that it satisfies the required conditions. Clearly $p''$ begins at $Z$, terminates at $Y$. The first segment $Z \overset{p'}{\dashrightarrow} M$ is active at non-colliders given $\mathrm{Anc}_{\lceil (\boldsymbol{X}(\mathcal{S}) \cup C_{\boldsymbol{X}(\mathcal{S}) \setminus Z}) \setminus Z \rceil}$ by the same argument as in Case 1, and at colliders by the definition of $M$. From $M \notin \mathrm{Anc}_{\lceil (\boldsymbol{X}(\mathcal{S}) \cup C_{\boldsymbol{X}(\mathcal{S}) \setminus Z}) \setminus Z \rceil}$, it follows that $M \dashrightarrow X' \dashrightarrow Y$ of $p''$ is active given $\mathrm{Anc}_{\lceil (\boldsymbol{X}(\mathcal{S}) \cup C_{\boldsymbol{X}(\mathcal{S}) \setminus Z}) \setminus Z \rceil}$, proving the result. $\square$

### B.2 Proof of Lemma 11

We begin by restating the lemma.

**Lemma 11.** *Let $\mathcal{G}(\mathcal{S})$ be a scoped graph that contains a context $Z_0 \in \boldsymbol{C}_{X_0}$ and satisfies the conditions of Theorem 8. Then, it contains the following:*

- *A **control path**: a directed path $d : A \dashrightarrow Z_0 \to X_0 \dashrightarrow Y$, where $A$ is a non-decision, possibly equal to $Z_0$, and $d$ contains no parents of $X_0$ other than $Z_0$.*
- *We can write $d$ as $A \dashrightarrow Z_{i_{min}} \to X_{i_{min}} \dashrightarrow \cdots Z_0 \to X_0 \dashrightarrow Z_{i_{max}} \to X_{i_{max}} \dashrightarrow Y, i_{min} \leq i \leq i_{max}$, where each $Z_i$ is the parent of $X_i$ along $d$ (where $A \dashrightarrow Z_{i_{min}}$ and $X_{i-1} \dashrightarrow Z_i$ are allowed to have length 0). Then, for each $i$, define the **info path**: $m_i' : Z_i \dashrightarrow Y$, active given $\lceil (\boldsymbol{X}(\mathcal{S}) \cup C_{\boldsymbol{X}(\mathcal{S}) \setminus Z_i}) \setminus Z_i \rceil$, that if $Z_i$ is a decision, begins as $Z_i \leftarrow N$ (so $N \in \boldsymbol{C}_{Z_i} \setminus \lceil (\boldsymbol{X}(\mathcal{S}) \cup C_{\boldsymbol{X}(\mathcal{S}) \setminus Z_i}) \setminus Z_i \rceil$.)*
- *Let $T_i$ be the node nearest $Y$ in $m_i' : Z_i \dashrightarrow Y$ (and possibly equal to $Z_i$) such that the segment $Z_i \overset{m_i'}{\dashrightarrow} T_i$ of $m_i'$ is identical to the segment $Z_i \overset{d}{\leftarrow} T_i$ of $d$. Then, let the **truncated info path** $m_i$ be the segment $T_i \overset{m_i'}{\dashrightarrow} Y$.*
- *Write $m_i$ as $m_i : T_i \dashrightarrow W_{i,1} \leftarrow U_{i,1} \dashrightarrow W_{i,2} \leftarrow U_{i,2} \cdots U_{i,J_i} \dashrightarrow Y$, where $J_i$ is the number of forks in $m_i$. (We allow the possibilities that $T_i = W_{i,1}$ so that $m_i$ begins as $T_i \leftarrow U_{i,1}$, or that $J_i = 0$ so that $m_i$ is $T_i \dashrightarrow Y$.) Then, for each $i$ and $1 \leq j \leq J_i$, let the **auxiliary path** be any directed path $r_{i,j} : W_{i,j} \dashrightarrow Y$ from $W_{i,j}$ to $Y$.*

The proof was described in Section 4.2.2, and is as follows.

*Proof.* We prove the existence of each path in turn.

From Lemma 9, there exists a control path $A \dashrightarrow Z_0$ that contains no parents of $X_0$ other than $Z_0$ (if $Z_0$ is a decision, choose $A = N$, and otherwise choose $A = Z_0$.) Moreover, from Theorem 8 condition (A), there exists a path $X_0 \dashrightarrow Y$, so we can concatenate these to obtain $d : A \dashrightarrow Z_0 \to X_0 \dashrightarrow Y$.

From condition (C) of Theorem 8, there exists an info path $m_i' : Z_i \dashrightarrow Y$, active given $\lceil (\boldsymbol{X}(\mathcal{S}) \cup C_{\boldsymbol{X}(\mathcal{S}) \setminus Z_i}) \setminus Z_i \rceil$, and if $Z_i$ is a decision, one that begins as $Z_i \leftarrow \cdot$, by Lemma 10. The existence of a truncated info path is immediate from this.

Each collider $W_{i,j}$ is an ancestor of $\lceil (\boldsymbol{X}(\mathcal{S}) \cup C_{\boldsymbol{X}(\mathcal{S}) \setminus Z_i}) \setminus Z_i \rceil$ by activeness, hence an ancestor of $\boldsymbol{X}(\mathcal{S}) \cup C_{\boldsymbol{X}(\mathcal{S}) \setminus Z_i}$ by the definition of the closure property $\lceil \cdot \rceil$, so $W_{i,j}$ is an ancestor of some $X \in \boldsymbol{X}(\mathcal{S})$; in addition,

from condition (A) of Theorem 8 we have $X \in \text{Anc}(Y)$. Hence, there exists a auxiliary path $r_{i,j} : W_{i,j} \dashrightarrow Y$. $\qquad\square$

### B.3 Proof of Lemma 13

We begin by proving an intermediate result.

**Lemma 27.** *Let* $\boldsymbol{w} = \langle w_0, \ldots, w_J \rangle$, $\bar{\boldsymbol{w}} = \langle \bar{w}_0, \ldots, \bar{w}_J \rangle$, *and* $\boldsymbol{u}_{0:J'} = \langle u_0, \ldots, u_{J'} \rangle$, $J' < J$ *where* $w_0, \bar{w}_0 \in \mathbb{B}^k$, $w_j, \bar{w}_j \in \mathbb{B}$ *for* $n \geq 1$, *and* $u_j \in \mathbb{B}^{\exp_2^n(k)}$. *If* $\boldsymbol{w}_{0:J'}$ *is consistent with* $\boldsymbol{u}_{0:J'}$ *but* $\bar{\boldsymbol{w}}_{0:J'}$ *is not compatible with* $u_{J'}$, *then there exists* $\boldsymbol{u} := \langle u_0, \ldots, u'_J, u_{J'+1}, \ldots, u_J \rangle$ *where* $u_j \in \mathbb{B}^{\exp_2^n(k)}$, *such that* $\boldsymbol{w}$ *is consistent with* $\boldsymbol{u}$, *but* $\bar{\boldsymbol{w}}$ *is not compatible with* $u_J$.

*Proof.* We will prove by induction. The base case $j = J' \geq 0$ is given by the condition.

Induction step: for $j > J'$, if $\boldsymbol{w}_{0:j-1} \sim \boldsymbol{u}_{0:j-1}$ and $\bar{\boldsymbol{w}}_{0:j-1} \not\sim u_{j-1}$, then there exists $\boldsymbol{u}_{0:j}$ such that (a) $\boldsymbol{w}_{0,j} \sim \boldsymbol{u}_{0:j}$ and (b) $\bar{w}_{0,j} \not\sim u_j$.

Let us construct $u_j$ such that $u_j[u_{j-1}] \leftarrow w_j$ and $u_j[i] \leftarrow 1 - \bar{w}_j$ for every $i \in \mathbb{B}^{\exp_2^{j-1}(k)} \setminus \{u_{j-1}\}$.

(a) First, by the construction $u_j[u_{j-1}] = w_j$ and given condition $\boldsymbol{w}_{0:j-1} \sim \boldsymbol{u}_{0:j-1}$, we can induce $\boldsymbol{w}_{0,j} \sim \boldsymbol{u}_{0,j}$.

(b) Next, we show that $\bar{\boldsymbol{w}}_{0:j} \not\sim u_j$. For the sake of contradiction, assume that $\bar{\boldsymbol{w}}_{0:j} \sim u_j$. Then, there exists $\boldsymbol{u}'_{0:j} = \langle u'_0, \ldots, u'_{j-1}, u_j \rangle$ satisfying $\bar{\boldsymbol{w}}_{0:j} \sim \boldsymbol{u}'_{0:j}$. Since $\bar{w}_{0,j-1} \not\sim u_{j-1}$, we can observe $u'_{j-1} \neq u_{j-1}$. Now, by construction, $u_j[u'_{j-1}] = 1 - \bar{w}_j \neq \bar{w}_j$. Thus, $\bar{\boldsymbol{w}}_{0:j-1} \not\sim u_j$. Contradiction.

By induction, $\bar{\boldsymbol{w}}$ is not compatible with $u_J$. $\qquad\square$

**Lemma 13.** *Let* $\boldsymbol{w} = \langle w_0, \ldots, w_J \rangle$ *and* $\bar{\boldsymbol{w}} = \langle \bar{w}_0, \ldots, \bar{w}_J \rangle$ *be sequences with* $w_0, \bar{w}_0 \in \mathbb{B}^k$, $w_j, \bar{w}_j \in \mathbb{B}$ *for* $j \geq 1$, *and let* $J' \leq J$ *be the smallest integer such that* $w_{J'} \neq \bar{w}_{J'}$. *Let* $u_0, \ldots, u_{J'}$ *be a sequence where* $u_j[u_{j-1}] = w_j$ *for* $1 \leq j < J'$. *Then, there exists some* $u_{J'+1}, \ldots, u_J$ *such that* $\boldsymbol{w}$ *is consistent with* $u_0, \ldots, u_J$, *but* $\bar{\boldsymbol{w}}$ *is incompatible with* $u_J$.

*Proof.* If $u_0, \ldots, u_n$ is incompatible with $\boldsymbol{w}$, then the result follows from Lemma 27. Otherwise, let $u_{n+1}$ be $w_n$ repeated $\exp_2^{n+1}(k)$ times. Then $u_0, \ldots, u_{n+1}$ is compatible with $w_0, \ldots, w_{n+1}$ but $u_{n+1}$ is incompatible with $\boldsymbol{b}$. We can then apply Lemma 27 to obtain the result. $\qquad\square$

### B.4 Proof of Lemma 17

We now prove the expected utility in the non-intervened model (which we will later establish is the MEU).

**Lemma 17.** *In the non-intervened model, the materiality SCM has* $Y = i_{max} - i_{min} + 1$, *surely.*

*Proof.* Since $Y = \sum_{i_{\min} \leq i \leq i_{\max}} Y^{m_i}$, it will suffice to prove that $Y^{m_i} = 1$ for every $i$. We will consider the cases where $m_i$ is, or is not, a directed path.

If the info path $m_i$ contains no collider, then every chain node $V$ in $d$ from $T_i$ to $Y$ has $V^d = \text{Pa}_V^d$, so $\text{pa}(Y^{\boldsymbol{p}_i}) = T_i^{\boldsymbol{p}_i}$. The same is true for the chain nodes in $m_i$, so $\text{Pa}^*(Y) = T_i^{\boldsymbol{p}_i}$, and so $Y^{m_i} = 1$, surely.

If $m_i$ contains a collider, each chain in $m_i$ and $r_{i,j}$ copies the value of its parent, so $\text{Pa}(Y^{\boldsymbol{p}_i, r_{i,1}, \ldots, r_{i,J^i}}) = \langle T_i^{\boldsymbol{p}_i}, W_{i,1}^{m_i}, \ldots, W_{i,J^i}^{m_i} \rangle$, and $\text{Pa}^*(Y) = U_{J^i}$. By construction, $\langle T_i^{\boldsymbol{p}_i}, W_{i,1}^{m_i}, \ldots, W_{i,J^i}^{m_i} \rangle$ is consistent with $\langle U_1, \ldots, U_{J^i} \rangle$, so by definition it is compatible with $U_{J^i}$, so $Y^{m_i} = 1$, surely. $\qquad\square$

### B.5 Proof of the requirements of an optimal policy

**Lemma 20** (Collider path requirement)**.** *If the materiality SCM has an info path* $m_i$ *that is not directed, and under the policy* $\boldsymbol{\pi}$ *there are assignments* $\text{Pa}(Y^{\boldsymbol{p}_i, r_{i,1:J_i}}) = \text{pa}(Y^{\boldsymbol{p}_i, r_{i,1:J_i}})$ *to parents of the outcome, and* $\boldsymbol{U}_{i,1:J_i}^{m_i} = \boldsymbol{u}_{i,1:J_i}^{m_i}$ *to the forks of* $m_i$, *with* $P^{\boldsymbol{\pi}}(\text{pa}(Y^{\boldsymbol{p}_i, r_{i,1:J_i}}), \boldsymbol{u}_{i,1:J_i}^{m_i}) > 0$ *and where* $\text{pa}(Y^{\boldsymbol{p}_i, r_{i,1:J_i}})$ *is inconsistent with* $\text{pa}(Y^{\boldsymbol{p}_i}), \boldsymbol{u}_{i,1:J_i}^{m_i}$, *then* $P^{\boldsymbol{\pi}}(Y^{m_i} < 1) > 0$.

*Proof of Lemma 20.* Let us index the forks and colliders of $m_i$ as $T_i \ \text{---} \ V_{i,1} \ \leftarrow\text{--} \ U_{i,1} \ \text{--}\rightarrow \ W_{i,1} \ \leftarrow\text{--}$ $,\ldots,W_{i,J^i} \ \leftarrow\text{--} \ U_{i,J^i} \ \text{--}\rightarrow \ Y$. Then, by assumption, there exists a set of assignments $\boldsymbol{w} := \mathrm{pa}(Y^{\boldsymbol{p}_i}), \boldsymbol{w}_{i,1:J_i}$ $\bar{\boldsymbol{w}} := \mathrm{pa}(Y^{\boldsymbol{p}_i}), \mathrm{pa}(Y^{\boldsymbol{r}_{i,1:J_i}})$ and $\boldsymbol{u} := \mathrm{pa}(Y^{\boldsymbol{p}_i}), \boldsymbol{u}_{i,1:J_i}^{m_i}$, where $\boldsymbol{w} \sim \boldsymbol{u}$ and $\bar{\boldsymbol{w}} \not\sim \boldsymbol{u}$ and $P^{\boldsymbol{\pi}}(\boldsymbol{w}, \bar{\boldsymbol{w}}, \boldsymbol{u}) > 0$. Let $J'$ be the smallest index such that $\bar{\boldsymbol{w}}_{1:J'} \not\sim \boldsymbol{u}_{1:J'}$, and clearly we will have $J' \geq 1$. Then, from Lemma 13, there exists $\bar{\boldsymbol{u}} = \mathrm{pa}(Y^{\boldsymbol{p}_i}), \boldsymbol{u}_{1:J'}, u_{i,J'+1}^{m_i}, \ldots, u_{i,J^i}^{m_i}$ such that $\boldsymbol{w} \sim \bar{\boldsymbol{u}}$ and $\bar{\boldsymbol{w}} \not\sim \bar{\boldsymbol{u}}_{J^i}$. Consider the intervention $\mathrm{do}(U_{i,J'+1}^{m_i}, \ldots, U_{i,J^i}^{m_i} = \bar{\boldsymbol{u}}_{J'+1:J^i})$. By the definition Definition 16, the intervention to forks on the info path can only affect variables outside of the info path via the intersection node $T_i$ and the colliders $W_{i,j}, 1 \leq j \leq J^i$. But $\bar{\boldsymbol{u}}_{1:J'} = \boldsymbol{u}_{1:J'}$, so $T_i$ and the colliders $W_{i,j}, 1 \leq j \leq J'$ are unchanged (note that this is true even if $T_i$ is a decision, which it can be). Furthermore, $\bar{\boldsymbol{w}} \sim \boldsymbol{u}$ so the colliders $W_{i,j}, J' < j \leq J^i$ are similarly unaffected by the intervention. We also have $\bar{\boldsymbol{w}} \not\sim \bar{\boldsymbol{u}}_{J^i}$. Then, by the same arguments as in the proof of Lemma 19, we have that $P^{\boldsymbol{\pi}}(Y^{m_i} = 0 \mid \mathrm{do}(\bar{\boldsymbol{u}})) > 1$ and then $P^{\boldsymbol{\pi}}(Y^{m_i} = 0) > 0$. $\square$

## B.6 Proof of Lemma 22

We begin by restating the lemma.

**Lemma 22** (Required properties unachievable if child is a non-decision). *Let $\mathcal{M}$ be a materiality SCM where the child of $X_0$ along $d$ is a non-decision. Then, the MEU for the scope $\mathcal{S}$ cannot be achieved by a deterministic policy in the scope $\mathcal{S}_{Z_0 \not\rightarrow X_0}$ (equal to $\mathcal{S}$, except that $Z_0$ is removed from $\boldsymbol{C}_{X_0}$).*

The proof was described in Section 4.4.1 and it is detailed as follows.

*Proof.* Consider the scope $\boldsymbol{X}(\mathcal{S})_{\backslash Z_0}$, equal to $\boldsymbol{X}(\mathcal{S})$ except that $\boldsymbol{C}_{X_0}$ is replaced with $\boldsymbol{C}_{X_0} \setminus \{Z_0\}$, and assume that a deterministic policy $\boldsymbol{\pi}$ in this scope achieves the MEU, then we will prove a contradiction. Specifically, we will establish two consequences that are clearly contradictory given a deterministic policy: (a) the support of $P^{\boldsymbol{\pi}}(X_0^{\boldsymbol{p}_0})$ contains at least $2^k$ assignments, (b) the domain of $\boldsymbol{C}_{X_0} \setminus \{Z_0\}$ contains fewer than $2^k$ assignments.

(Proof of a.) We know that $A$ assigns a strictly positive probability to $2^k$ assignments (Definition 16) and so if $\boldsymbol{\pi}$ achieves the MEU, then $\mathrm{Pa}(Y^d) \stackrel{\mathrm{a.s.}}{=\!=} A$ (Lemma 21). So $\mathrm{Pa}(Y^d)$ has at least $2^k$ assignments in its support. Let us now consider the cases where $X_0$ is, or is not, the decision nearest $Y$ along $d$.

If $X_0$ is the decision nearest $Y$ along $d$, then by the model definition, $\mathrm{Pa}(Y^d) = X_0^d$ surely, so $X_0$ must have at least $2^k$ assignments in its support, and so (a) follows.

If $X_0$ is not the decision nearest $Y$ along $d$, then note that by assumption, there are one or more chance nodes in $d$ separating $X_0$ from $X_1$. Furthermore, $T_1$ must be one of these nodes (because $T_1$ is defined by a segment $T_1 \ \text{--}\rightarrow \ Z_1$, shared by $d$ and $m_i'$, and active given $\lceil (\boldsymbol{X}(\mathcal{S}) \cup C_{\boldsymbol{X}(\mathcal{S}) \backslash Z_1}) \setminus Z_1 \rceil$, and such a path cannot be active if it includes $X_0$.) The materiality SCM is constructed to pass values along $d$, and since the segment $T_1 \ \text{--}\rightarrow \ Z_1$ has no decisions, we have $T_1^d = X_0^d$, surely. Since $T_1$ is a chance node, if $\boldsymbol{\pi}$ achieves the MEU, we also have by Lemma 18 and Lemma 19 that $\mathrm{Pa}(Y^{\boldsymbol{p}_1}) \stackrel{\mathrm{a.s.}}{=\!=} T_1^{\boldsymbol{p}_1}$ and, since $d \in \boldsymbol{p}_1$, that $\mathrm{Pa}(Y^d) \stackrel{\mathrm{a.s.}}{=\!=} T_1^d$. So $X_0^d \stackrel{\mathrm{a.s.}}{=\!=} \mathrm{Pa}(Y^d)$. Since $\mathrm{Pa}(Y^d)$ places strictly positive probability on at least $2^k$ assignments, so does $X_0^d$.

(Proof of b.) The domain of $\boldsymbol{C}_{X_0} \setminus \{Z_0\}$ is a Cartesian product of variables $V^p$ for $V \in \boldsymbol{C}_{X_0} \setminus \{Z_0\}$ where $p$ is either $d$, some $m_i$ or some $r_{i,j}$ Definition 16.

The control path $d$ does not intersect $\boldsymbol{C}_{X_0} \setminus \{Z_0\}$ as it is defined not to include parents of $X_0$ other than $Z_0$ (Lemma 11). Each info path $m_i$ is active given $\lceil (\boldsymbol{X}(\mathcal{S}) \cup C_{\boldsymbol{X}(\mathcal{S}) \backslash Z_0}) \setminus Z_0 \rceil$ (Lemma 11), so can only intersect $\boldsymbol{C}_{X_0} \setminus \{Z_0\}$ at the colliders, which have domain $\mathbb{B}$. Finally, any variable in a path $r_{i,j}$ would also have domain $\mathbb{B}$. So the domain of $\boldsymbol{C}_{X_0} \setminus Z_0$ is not larger than $2^{c \cdot |\boldsymbol{C}_{X_0}|}$, where $c$ is the maximum number of materiality paths passing through any vertex in the graph, and $|\boldsymbol{C}_{X_0}|$ is the number of variables in $\boldsymbol{C}_{X_0}$. By construction, $k > c \cdot \max_{X \in \boldsymbol{X}(\mathcal{S})} |C_X|$, so the domain of $\boldsymbol{C}_{X_0} \setminus Z_0$ is less than $2^k$, proving (b).

A deterministic policy cannot map fewer than $2^k$ assignments to greater than $2^k$ assignments, and so (a-b) imply a contradiction. $\square$

## B.7 Proof of Lemma 23

We firstly restate the lemma.

**Lemma 23** (If next fork is repeated, then fork only influences intersection node). *If, in the materiality SCM:*

- *the intersection node $T_i$ is the vertex $X_{i-1}$,*
- *$\pi_{T_i}$ is a deterministic decision rule where $\pi_{T_i}(\boldsymbol{c}^{\neg m_i}(T_i, u_{i,1}) = \pi_{T_i}(\boldsymbol{c}^{\neg m_i}(T_i, u'_{i,1}))$ for assignments $u_{i,1}, u'_{i,1}$ to the first fork variable, and $\boldsymbol{c}^{\neg m_i}(T_i)$ to the contexts of $T_i$ not on $m_i$, and*
- *$\boldsymbol{W}_{i,1:J_i} = \boldsymbol{w}_{i,1:J_i}$, and $\boldsymbol{U}_{i,2:J_i} = \boldsymbol{u}_{i,2:J_i}$ are assignments to forks and colliders in $m_i$ where each $u_{i,j}$ consists of just $w_{i,j}$ repeated $\exp_2^j(k + |\boldsymbol{p}_i| - 1)$ times, then:*

$$P^{\boldsymbol{\pi}}(pa(Y^{\boldsymbol{p}_i, r_{i,1}}), \boldsymbol{c}^{\neg m_i}(T_i), \boldsymbol{w}_{i,1:J_i}, \boldsymbol{u}_{i,2:J_i} \mid \mathrm{do}(u_{i,1})) = P^{\boldsymbol{\pi}}(pa(Y^{\boldsymbol{p}_i, r_{i,1}}), \boldsymbol{c}^{\neg m_i}(T_i), \boldsymbol{w}_{i,1:J_i}, \boldsymbol{u}_{i,2:J_i} \mid \mathrm{do}(u'_{i,1})).$$

The proof is as follows.

*Proof.* An intervention $\mathrm{do}(u'_{i,1})$ could, in the materiality SCM (Definition 16) only affect the variables $\mathrm{Pa}(Y^{\boldsymbol{p}_i, r_{i,1}}), \boldsymbol{C}^{\neg m_i}(T_i), \boldsymbol{W}_{i,1:J_i}, \boldsymbol{U}_{i,2:J_i}$ in four ways:

1. via the intersection node $T_i$,
2. via the collider $W_{i,2}$ of $m_i$,
3. via contexts lying in the segment $m_i : T_i \dashleftarrow U_{i,1} \dashrightarrow W_{i,2}$,
4. if $\mathrm{Pa}_Y^{\boldsymbol{p}_i}, \boldsymbol{C}^{\neg m_i}(T_i)$ or $\boldsymbol{U}_{i,2:J_i}$ were distinct from $T_i, W_{i,2}$ and lay on $m_i : T_i \dashleftarrow U_{i,1} \dashrightarrow W_{i,2}$

The deterministic decision rule has $\boldsymbol{\pi}_{T_i}(u_{i,1}, \boldsymbol{c}^{\neg m_i}(T_i)) = \boldsymbol{\pi}_{T_i}(u'_{i,1}, \boldsymbol{c}^{\neg m_i}(T_i))$, so (1) is false. Also, $u_{i,2}$ equals $w_{i,2}$ repeated, so $u_{i,2}[x] = w_{i,2}$ for all $x$, and thus (2) is false also. Moreover, $m_i : T_i \dashleftarrow U_{i,1} \dashrightarrow W_{i,2}$ is active given $\lceil (\boldsymbol{X}(\mathcal{S}) \cup C_{\boldsymbol{X}(\mathcal{S}) \setminus T_i}) \setminus T_i \rceil$ and so contexts can only lie at the endpoints $T_i$ and the collider $W_{i,2}$, meaning that (3) is false. Finally, $\mathrm{Pa}_Y^{\boldsymbol{p}_i}$ is a descendant of $T_i$ by the definition of the control path, so can only lie on $m_i : T_i \dashleftarrow U_{i,1} \dashrightarrow W_{i,2}$ if it is the vertex $T_i$, which we have already proved is not influenced by $u_{i,1}$; meanwhile, $\boldsymbol{C}^{\neg m_i}(T_i)$ does not intersect $m_i$ by definition, and $\boldsymbol{U}_{i,2:J_i}$ are fork variables, which cannot lie on $m_i : T_i \dashleftarrow U_{i,1} \dashrightarrow W_{i,2}$, so (4) is false, and the result follows. $\square$

## B.8 Proof of Lemma 24

We begin by restating the lemma.

**Lemma 24** (Decision must distinguish fork values). *If in the materiality SCM:*

- *the intersection node $T_i$ is the vertex $X_{i-1}$, and*
- *$\boldsymbol{\pi}$ is a deterministic policy that for assignments $u_{i,1}, u'_{i,1}$ to $U_{i,1}$ where $u_{i,1} \neq u'_{i,1}$,*      (†)
  *has $\pi_{T_i}(\boldsymbol{c}^{\neg m_i}(T_i), u_{i,1}) = \pi_{T_i}(\boldsymbol{c}^{\neg m_i}(T_i), u'_{i,1})$ for every $\boldsymbol{C}_{T_i}^{\neg m_i}(T_i) = \boldsymbol{c}^{\neg m_i}(T_i)$,*

*then $P^{\boldsymbol{\pi}}(Y^{m_i} < 1) > 0$*

The proof has been described already, and it proceeds as follows.

*Proof.* Let us assume Equation (†), and that the MEU is achieved, and we will prove a contradiction. Given Equation (†), there is an index at which $u_{i,1}$ and $u'_{i,1}$ differ. We write this index as an assignment $\mathrm{pa}(Y^{d, r_{i,j}})$, belonging to $\mathrm{Pa}(Y^{\boldsymbol{p}_i})$. Define each $u_{i,j}, 2 \leq j \leq J_i$ as equal to $\mathrm{pa}(Y^{r_{i,j}})$, repeated $\exp_2^j(k + |\boldsymbol{p}_i| - 1)$ times. Then, we have:

$$0 < P^{\boldsymbol{\pi}}(A^d = \mathrm{pa}(Y^d), \boldsymbol{U}_{i,1:J_i} = \boldsymbol{u}_{i,1:J_i})$$

because $A$ and $\boldsymbol{U}_{i,1:J_i}$ are independent random variables with full support. Then, let $\boldsymbol{c}^{\neg m_i}(T_i)$ and $\boldsymbol{w}_{1,1:J_i}$ be any assignments to the parents of $T_i$ not on $m_i$, and to the colliders on $m_i$ such that:

$$0 < P^{\boldsymbol{\pi}}(A^d = \mathrm{pa}(Y^d), \boldsymbol{c}^{\neg m_i}(T_i), \boldsymbol{w}_{1,1:J_i}, \boldsymbol{u}_{i,1:J_i}).$$

Given these assignments, in order to achieve $P^{\boldsymbol{\pi}}(Y^{m_i} = 1) = 1$, we must have $\mathrm{Pa}(Y^d) \overset{\text{a.s.}}{=\!=} A^d$ (Lemma 21) and $\mathrm{pa}(Y^{\boldsymbol{p}_i})$ must be consistent with $\boldsymbol{u}_{i,1:J_i}$ (Lemma 20). We must also therefore have $\mathrm{Pa}(Y^{\boldsymbol{p}_i,\boldsymbol{r}_{i,1:J_i}}) = \mathrm{pa}^{\boldsymbol{p}_i,\boldsymbol{r}_{i,1:J_i}})$, so marginalizing over $A^d$, we must have:

$$
\begin{aligned}
0 &< P^{\boldsymbol{\pi}}(\mathrm{Pa}(Y^{\boldsymbol{p}_i,\boldsymbol{r}_{i,1:J_i}}) = \mathrm{pa}(Y^{\boldsymbol{p}_i,\boldsymbol{r}_{i,1:J_i}}), \boldsymbol{c}^{\neg m_i}(T_i), \boldsymbol{w}_{1,1:J_i}, \boldsymbol{u}_{i,1:J_i}) \\
\therefore 0 &< P^{\boldsymbol{\pi}}(\mathrm{pa}(Y^{\boldsymbol{p}_i,\boldsymbol{r}_{i,1:J_i}}), \boldsymbol{c}^{\neg m_i}(T_i), \boldsymbol{w}_{1,1:J_i}, \boldsymbol{u}_{i,2:J_i} \mid \mathrm{do}(u_{i,1})) &&(U_{i,1:J_i} \text{ unconfounded}) \\
&= P^{\boldsymbol{\pi}}(\mathrm{pa}(Y^{\boldsymbol{p}_i,\boldsymbol{r}_{i,1:J_i}}), \boldsymbol{c}^{\neg m_i}(T_i), \boldsymbol{w}_{1,1:J_i}, \boldsymbol{u}_{i,2:J_i} \mid \mathrm{do}(u'_{i,1})) &&(\text{by } Lemma\ 23) \\
&= P^{\boldsymbol{\pi}}(\mathrm{pa}(Y^{\boldsymbol{p}_i,\boldsymbol{r}_{i,1:J_i}}), \boldsymbol{c}^{\neg m_i}(T_i), \boldsymbol{w}_{1,1:J_i}, \boldsymbol{u}_{i,2:J_i} \mid u'_{i,1}) &&(P^{\boldsymbol{\pi}}(u'_{i,1}) > 0.) \\
\therefore 0 &< P^{\boldsymbol{\pi}}(\mathrm{pa}(Y^{\boldsymbol{p}_i,\boldsymbol{r}_{i,1:J_i}}), u'_{i,1}) &&(P^{\boldsymbol{\pi}}(u'_{i,1}) > 0.)
\end{aligned}
$$

However, $u'_{i,1}[\mathrm{pa}(Y^{\boldsymbol{p}_i})] \neq u_{i,1}[\mathrm{pa}(Y^{\boldsymbol{p}_i})]$ and $u_{i,1}[\mathrm{pa}(Y^{\boldsymbol{p}_i})] = \mathrm{pa}(Y^{r_{i,1}})$, so $\mathrm{pa}(Y^{\boldsymbol{p}_i}), \mathrm{pa}(Y^{\boldsymbol{r}_{i,1:J_i}})$ is inconsistent with $\mathrm{pa}(Y^{\boldsymbol{p}_i}), u'_{i,1}, \boldsymbol{u}_{i,2:J_i}$. So $0 < P^{\boldsymbol{\pi}}(\mathrm{pa}(Y^{\boldsymbol{p}_i,\boldsymbol{r}_{i,1:J_i}}), u'_{i,1})$ implies that $P^{\boldsymbol{\pi}}(Y_1 = 1) < 1$ (by Lemma 20), and the MEU is not achieved. $\qquad\square$

## C  Proof of Lemma 25

We first restate the lemma.

**Lemma 25** (Required properties unachievable if child is a decision)**.** *Let $\mathcal{M}$ be the materiality SCM for some scoped graph $\mathcal{G}_{\mathcal{S}}$, where $i_{max} > 0$ and $T_1$ is a decision. Then, there exists no deterministic policy in the scope $\mathcal{S}_{Z_0 \not\to X_0}$ that achieves the MEU.*

The proof was explained in section Section 4.4.2, and is detailed as follows.

*Proof.* To begin with, by assumption, the child of $X_0$ along $d$ is a decision, so $X_0$ is the same node as $Z_1$, and since the segment $T_1 \dashrightarrow X_1$ must be active given $\lceil (\boldsymbol{X}(\mathcal{S}) \cup C_{\boldsymbol{X}(\mathcal{S})\setminus Z_1}) \setminus Z_1 \rceil$, $X_0$ is also $T_0$. We will now bound the domains of $X_0$ and $\boldsymbol{C}^{\neg m_1}(X_0)$.

*The domain of $X_0$.* Given that $X_0$ is a decision, while each truncated info path $m_{i'}$ is active given $\lceil (\boldsymbol{X}(\mathcal{S}) \cup C_{\boldsymbol{X}(\mathcal{S})\setminus Z_i}) \setminus Z_i \rceil$, it follows that $X_0$ cannot overlap with info paths, except for colliders of $m_{i'}, i' \neq i$, and the endpoint of $m_1$. As such, the domain of $T_0$ is at most $|\mathfrak{X}_{X_0}| \leq 2^{k+c}$, due to $k$ bits from $d$ (Definition 16), and at most $c$ bits from the info paths and auxiliary paths (where $c$ is the maximum number of materiality paths passing through any vertex in the graph).

*The domain of $\boldsymbol{C}^{\neg m_1}(X_0)$.* Given that each info path $m_i$ is active given $\lceil (\boldsymbol{X}(\mathcal{S}) \cup C_{\boldsymbol{X}(\mathcal{S})\setminus Z_i}) \setminus Z_i \rceil$, the contexts $\boldsymbol{C}^{\neg m_1}(X_0)$ cannot intersect any $m_i$, except at colliders in $m_i$. Moreover, by the definition of the control path, the only parent of $X_0$ that it contains is $Z_0$. So, $\boldsymbol{C}^{\neg m_1}(X_0)$ can only intersect portions of the materiality paths with domain $\mathbb{B}$, and so the size of the domain of $\boldsymbol{C}^{\neg m_1}(X_0)$ cannot exceed $|\mathfrak{X}_{\boldsymbol{C}^{\neg m_1}(X_0)\setminus Z_0}| \leq 2^{bc}$, where $b$ is the maximum number of variables belonging to any context $C_X$, and $c$ is the largest number of materiality paths passing through any vertex.

*Proof of Equation (†)* As the domain of $X_0$ has $\mathfrak{X}_{X_0}| \leq 2^{k+c}$, for any particular $\boldsymbol{C}^{\neg m_1}(X_0) = \boldsymbol{c}^{\neg m_1}(X_0)$, there are at most $2^{k+c}$ assignments $\mathfrak{X}_{U'_{1,1}} \subseteq \mathfrak{X}_{U_{1,1}}$ such that for all $u_{1,1}, u'_{1,1} \in \mathfrak{X}_{U'_{1,1}}$, $\pi_{X_1}(\boldsymbol{c}^{\neg m_1}(X_0), u_{1,1}) \neq \pi_{X_1}(\boldsymbol{c}^{\neg m_1}(X_0), u'_{1,1})$. Furthermore, as $|\mathfrak{X}_{\boldsymbol{C}^{\neg m_1}(X_0)\setminus Z_0}| \leq 2^{bc}$, by the union property, there are at most $2^{bc(k+c)}$ assignments $\mathfrak{X}'_{U_{1,1}}$ such that there exists $\boldsymbol{c}^{\neg m_1}(X_0)$ such that for all $u_{1,1}, u'_{1,1} \in \mathfrak{X}_{U_{1,1}}$, $u_{1,1}, u'_{1,1} \in \mathfrak{X}_{U'_{1,1}}$, $\pi_{X_1}(\boldsymbol{c}^{\neg m_1}(X_0), u_{1,1}) = \pi_{X_1}(\boldsymbol{c}^{\neg m_1}(X_0), u'_{1,1})$. However, the domain of $U_i$ is $\mathbb{B}^{\exp_2^1(k+|\boldsymbol{p}_0|-1)} \supseteq \mathbb{B}^{2^k}$ (as $\boldsymbol{p}_0$ contains at least $d$), so:

$$|\mathfrak{X}_{\mathrm{Pa}(X_0^{m_i})}| \geq 2^{2^k} > 2^{(k+c)bc} \geq |\mathfrak{X}_{\boldsymbol{C}^{\neg m_1}(X_0)}||\mathfrak{X}_{X_0}|,$$

where the strict inequality is from the definition of $k$ in Definition 16. So, there must exist a pair of assignments $u_{1,1}, u'_{1,1}$ in the domain of $U_{1,1}$ such that for all $\boldsymbol{c}^{\neg m_1}(X_0) \in \mathfrak{X}_{\boldsymbol{C}^{\neg m_1}(X_0)}$, $\pi_{X_1}(\boldsymbol{c}^{\neg m_1}(X_0), u_{1,1}) = \pi_{X_1}(\boldsymbol{c}^{\neg m_1}(X_0), u'_{1,1})$. This satisfies Equation (†), which by Lemma 24 proves the result. $\qquad\square$

# D  Supplementary proofs for Section 5 (Proof of Lemma 26)

## D.1  Proving the existence of paths

In this section, we will prove that when LB-factorizability is not satisfied, then there exist info paths and control paths, a potential intermediate step toward establishing completeness of Theorem LB-2 from Lee & Bareinboim (2020).

**Lemma 26** (System Exists General). *Let $\mathcal{G}_S$ be a scoped graph that satisfies conditions (A,B) from Theorem 8. If $\boldsymbol{Z} = \{Z\}$, $\boldsymbol{X}' \supseteq Ch(Z)$, $\boldsymbol{C}' = C_{\boldsymbol{X}'} \setminus (\boldsymbol{X}' \cup \boldsymbol{Z})$, $\boldsymbol{U} = \emptyset$ are not LB-factorizable, then there exists a pair of paths to some $C' \in \boldsymbol{C}' \cup Y$:*

- *an info path $m : Z \dashrightarrow C'$, active given $\lceil \boldsymbol{X}' \cup \boldsymbol{C}' \rceil$, and*
- *a control path $d : X \dashrightarrow C'$ where $X \in \boldsymbol{X}'$.*

Since we will have to establish activeness given a set of implied variables, the following lemma will be useful.

**Lemma 28.** *Let $p$ be a path. If (i) $p$ contains no non-collider in $\boldsymbol{N}$, (ii) every fork variable in $p$ is not in $\lceil \boldsymbol{N} \rceil$, and (iii) every endpoint of $p$ that has a child along $p$ is not in $\lceil \boldsymbol{N} \rceil$, then $p$ contains no non-collider in $\lceil \boldsymbol{N} \rceil$.*

*Proof.* Write $p$ as $W_1 \leftarrow\!\!- U_1 -\!\!\dashrightarrow W_2 \leftarrow\!\!- U_2 \ldots U_J -\!\!\dashrightarrow W_{J+1}$, where possibly $W_1$ is $U_1$, and possibly $U_J$ is $W_{J+1}$. Every $U_j$ is not in $\lceil \boldsymbol{N} \rceil$ by (ii-iii). Each non-collider child $V$ of any $U_j$ has a parent that is not in $\lceil \boldsymbol{N} \rceil$, and $V \notin \boldsymbol{N}$ by (i), so $V \notin \lceil \boldsymbol{N} \rceil$. The same is then true for the non-collider child of $V$, and so on. Since every non-collider $V'$ in $p$ has a segment $U_j -\!\!\dashrightarrow V'$ of $p$ consisting of only non-colliders, every $V' \notin \lceil \boldsymbol{N} \rceil$, and $V'$ contains no non-collider in $\lceil \boldsymbol{N} \rceil$, proving the result. $\qquad\square$

Conditions II-III of LB-factorizability require that there must exist an ordering over variables, that where certain variables are placed before others (i.e. that satisfies certain precedence relationships). Our approach will be to encode the precedence relationships from condition III in a graph, as follows.

**Definition 29.** Let the "ordering graph" $\mathcal{H}$ be a graph on vertices $\boldsymbol{Z} \cup \boldsymbol{X}' \cup \boldsymbol{C}'$, with an edge $A \to B$ from each parent $A \in \mathrm{Pa}(B)$ of a decision $B \in \boldsymbol{X}'$, and an edge $B \to C$ from each decision $B \in \boldsymbol{X}'$ to a descendant $C \in \mathrm{Desc}(B)$.

A useful property of the ordering graph is that if a variable $V$ is downstream of a context $C$ in the ordering graph, then there exists a decision, that has $C$ as a context, and can influence $V$.

**Lemma 30.** *If vertex $V$ is a descendant in $\mathcal{H}$ of a context $Z \in C_{\mathcal{S}(\boldsymbol{X})}$, then $\mathcal{G}_S$ contains a path $Z \to X \dashrightarrow V$, where $X \in \boldsymbol{X}'$.*

*Proof.* Assume that $V \in \mathrm{Desc}^{\mathcal{H}}(Z)$. The path in $\mathcal{H}$ from $Z$ begins with an edge $Z \to X$ where $X \in \boldsymbol{X}'$, which implies that $\mathcal{G}_S$ has an edge $Z \to X$. The path in $\mathcal{H}$ must continue from $X$ to $Z$, and since each edge $A \to B$ in $\mathcal{H}$ has $B \in \mathrm{Desc}^{\mathcal{G}_S}(A)$, it follows that $V \in \mathrm{Desc}^{\mathcal{G}_S}(X)$, proving the result. $\qquad\square$

It is also useful to note that the expression $\boldsymbol{\pi}_{\boldsymbol{X}'_{\prec C}}$ is unnecessary in condition II.

**Lemma 31** (Unnecessary separation in condition II). *Let $\boldsymbol{X}'$ be a set of decisions, $\boldsymbol{Z}$ be a set of variables disjoint with $\boldsymbol{X}'$, and $\boldsymbol{C}'$ be the set of contexts not in $\boldsymbol{C}'$ or $\boldsymbol{Z}$, and $\prec$ be an ordering over $\boldsymbol{C}' \cup \boldsymbol{X}' \cup \boldsymbol{Z}$. If $\boldsymbol{\pi}_{\boldsymbol{X}'_{\prec C}} \not\perp C \mid \lceil (\boldsymbol{X}' \cup \boldsymbol{C}')_{\prec C} \rceil$ for some $C \in \boldsymbol{C}'$ then $\boldsymbol{Z}_{\prec C} \not\perp C \mid \lceil (\boldsymbol{X}' \cup \boldsymbol{C}')_{\prec C} \rceil$*

*Proof.* By assumption, there is a path $p$ from $\boldsymbol{\pi}_X$ to $C$, active given $\lceil (\boldsymbol{X}' \cup \boldsymbol{C}')_{\prec C} \rceil$, for some $X \in \boldsymbol{X}'_{\prec C}$. The only neighbour of $\pi_X$ is $X$, so $p$ must terminate as $X \leftarrow \pi_X$. As $X$ is in $\boldsymbol{X}'$, activeness given $\lceil (\boldsymbol{X}' \cup \boldsymbol{C}')_{\prec C} \rceil$ implies that $p$ terminates as $C \to X \leftarrow \pi_X$. Every parent of $X$ is in $\boldsymbol{X}' \cup \boldsymbol{C}'$ except $\boldsymbol{Z}$. So by truncating $p$ at $\boldsymbol{Z}$, we have that there is a path from $\boldsymbol{Z}_{\prec C}$ to $C$, active given $\lceil (\boldsymbol{X}' \cup \boldsymbol{C}')_{\prec C} \rceil$. $\qquad\square$

We are now equipped to prove Lemma 26. Recall that for $\boldsymbol{Z}, \boldsymbol{X}'$ to be LB-factorizable, there only needs to be one ordering $\prec$ that satisfies the precedence relationships from conditions II-III. So the approach in our proof will be to define one such $\prec$ that satisfies conditions III. Since $\boldsymbol{Z}, \boldsymbol{X}'$ are not LB-factorizable, that must mean that condition I or II is violated, which will imply the existence of paths $m, d$ in each case. (We will use the notation $\mathrm{Desc}^{\mathcal{H}}(Z_0)$ to denote the set of vertices that are descendants of $Z_0$ in the ordering graph $\mathcal{H}$.)

*Proof of Lemma 26.* Let $\prec$ be any ordering $\langle V_0, \cdots V_m, Z_0, V_{m+2}, \cdots V_M \rangle$, over $\boldsymbol{Z} \cup \boldsymbol{X}' \cup \boldsymbol{C}$ that is topological in $\mathcal{H}$ and where $V_{m+2}, \cdots, V_M$ are in $\mathrm{Desc}^{\mathcal{H}}(Z_0)$ whereas $V_0 \cdots V_m$ are not. Since $\prec$ is topological in $\mathcal{H}$, Condition III is satisfied, and since LB factorizability is not satisfied, Condition I or II must be be violated; we consider these cases in turn.

**Case 1: Condition I is violated.**

If Condition I is violated, there is a path $m' : V_1, V_2, \cdots, V_n$ where $V_1 = \boldsymbol{\pi}_{\boldsymbol{X}'}$ and $V_n = Y$, active given $\lceil \boldsymbol{X}' \cup \boldsymbol{C}' \rceil$. From the definition of $\boldsymbol{\pi}_X$, this path must begin as $\boldsymbol{\pi}_{\boldsymbol{X}'} \to X$ for $X \in \boldsymbol{X}'$. As $X$ is in the conditioning set, it must be a collider, i.e. $m'$ begins as $\Pi_X \to X \leftarrow V_3$. The only parent of $X$ that is not in the conditioning set is $Z_0$, so we have $\Pi_X \to X \leftarrow Z_0 \text{ --- } Y$. We truncate $m'$ as $m : Z_0 \text{ --- } Y$. Since $Z_0 \to X$ satisfies condition (A) of Theorem 8, there exists some $d : X \dashrightarrow Y$, proving the result in this case.

**Case 2: Condition II is violated. Step 2.1**

The violation of condition II implies that there is an active path from some $C \in \boldsymbol{C}'$ to $\pi_{\boldsymbol{X}'_{\prec C}}, \boldsymbol{Z}_{\prec C}$, or $\boldsymbol{U}'$. This path cannot go to $\boldsymbol{U}'$, which was chosen to be empty. Moreover, if there is an active path to $\pi_{\boldsymbol{X}'_{\prec C}}$, then there is a similarly active path to $\boldsymbol{Z}_{\prec C}$ (Lemma 31. So let $m' : Z_0 \text{ --- } C'$ (where $Z_0 \prec C'$) be the path to $Z_0$, active given $\lceil (\boldsymbol{X}' \cup \boldsymbol{C}')_{\prec C} \rceil$. Replace this path with a walk $w'$ with an added segment $V \dashrightarrow S \dashleftarrow V$ from each collider $Z$ to a variable $S$ in the conditioning set. Truncate $w'$ as $Z_0 \text{ --- } C$, where $C$ is the node in $\boldsymbol{C}'_{\succ Z_0}$ nearest $Z_0$ along $w'$. Then let $m$ be the path obtained from $w$ by removing all retracing segments. Clearly $m$ is active given $\lceil (\boldsymbol{X}' \cup \boldsymbol{C}')_{\prec C} \rceil$ From $Z_0 \prec C$, it follows that $C \in \mathrm{Desc}^{\mathcal{H}}(Z_0)$, so there exists a path $d : Z_0 \to X \dashrightarrow C$ for $X \in \boldsymbol{X}'$ (Lemma 30).

**Case 2: Condition II is violated. Step 2.2**

We will now establish that $m$ is active given $\lceil \boldsymbol{X}' \cup \boldsymbol{C}' \rceil$. Since $m$ is active given $\lceil (\boldsymbol{X}' \cup \boldsymbol{C}')_{\prec C} \rceil$, and $\lceil (\boldsymbol{X}' \cup \boldsymbol{C}') \rceil \supseteq \lceil (\boldsymbol{X}' \cup \boldsymbol{C}')_{\prec} C \rceil$, $m$ is active given $\lceil \boldsymbol{X}' \cup \boldsymbol{C}' \rceil$ at each collider. We now prove that $m$ also contains no non-collider in $\lceil (\boldsymbol{X}' \cup \boldsymbol{C}')_{\prec C} \rceil$ using Lemma 28, by proving that the non-colliders are not in $(\boldsymbol{X}' \cup \boldsymbol{C}')$ while the endpoints and forks are not in $\lceil (\boldsymbol{X}' \cup \boldsymbol{C}') \rceil$.

*Step 2.2.1: no non-collider in $w$ is in $(\boldsymbol{X}' \cup \boldsymbol{C}')$.*

We consider three sub-cases: a non-collider in 2.2.1.1: $(\boldsymbol{C}' \cup \boldsymbol{X}')_{\prec C}$, 2.2.1.2: $\boldsymbol{C}'_{\succ C}$, or 2.2.1.3: $\boldsymbol{X}'_{\succ C}$. *Sub-case 2.2.1.1: a non-collider in $(\boldsymbol{C}' \cup \boldsymbol{X}')_{\prec C}$.* As $w$ is active given $\lceil (\boldsymbol{X}' \cup \boldsymbol{C}')_{\prec} C \rceil$, $w$ does not contain a non-collider in $(\boldsymbol{C}' \cup \boldsymbol{X}')_{\prec C}$. *Sub-case 2.2.1.2: a non-collider in $\boldsymbol{C}'_{\succ C}$.* Moreover, the definition of $C$ implies that $m$ cannot contain a non-collider in $\boldsymbol{C}'_{\succ C}$. *Sub-case 2.2.1.3: a non-collider in $\boldsymbol{X}'_{\succ C}$.* Finally, $w$ cannot contain any non-collider $X \in \boldsymbol{X}'_{\succ C}$, because being a vertex being a non-collider in any path implies that it is an ancestor of a collider or an endpoint of that path, but being an ancestor of a collider or an endpoint of $w$ implies $X \prec C$, which is a contradiction. If $X$ is an ancestor of the endpoint $C$, then by the definition of $\mathcal{H}$, $X \prec C$, which contradicts $X \in \boldsymbol{X}'_{\succ C}$. If $X$ is an ancestor of the other endpoint $Z_0$, then $X \prec Z_0$ by the definition of $\mathcal{H}$, and so $X \prec C$, implying a contradiction once again. If $X$ is an ancestor of a collider $V$, then by activeness, the collider must have a descendant $V'$ in $\lceil (\boldsymbol{X}' \cup \boldsymbol{C}')_{\prec C} \rceil$, and so $X$ is an ancestor of $V'$. By the definition of $\mathcal{H}$, it follows that $X \prec V'$, and since $V' \prec C$, we have $X \prec C$. Since no non-collider in $w$ is in $(\boldsymbol{X}' \cup \boldsymbol{C}')$, it also follows that no non-collider in $m$ is in $(\boldsymbol{X}' \cup \boldsymbol{C}')$.

*Step 2.2.2: no endpoint of $m$ is in $\lceil (\boldsymbol{X}' \cup \boldsymbol{C}') \rceil$.*

The endpoint $Z_0$ cannot be in $\lceil (\boldsymbol{X}' \cup \boldsymbol{C}')_{\prec C} \rceil$ because $Z_0 \in \boldsymbol{Z}$, and $\boldsymbol{Z}$ is disjoint from $\boldsymbol{X}'$ and $\boldsymbol{C}'$. The endpoint $C$ cannot be in $\lceil (\boldsymbol{X}' \cup \boldsymbol{C}')_{\prec C} \rceil$ because we cannot have $C \prec C$.

*Step 2.2.3: If no non-collider in $(\boldsymbol{X}' \cup \boldsymbol{C}')$ then no fork in $\lceil \boldsymbol{X}' \cup \boldsymbol{C}' \rceil \setminus \boldsymbol{X}' \cup \boldsymbol{C}'$.*

Assume that a fork $V$ in $\lceil \boldsymbol{X}' \cup \boldsymbol{C}' \rceil \setminus \boldsymbol{X}' \cup \boldsymbol{C}'$ is in $m$, and we will prove a contradiction. The vertex $V$ must not be in $\lceil (\boldsymbol{X}' \cup \boldsymbol{C}')_{\prec C} \rceil$, since $m'$ is active given $\lceil (\boldsymbol{X}' \cup \boldsymbol{C}')_{\prec C} \rceil$. As $V$ is in $\lceil \boldsymbol{X}' \cup \boldsymbol{C}' \rceil \setminus \lceil (\boldsymbol{X}' \cup \boldsymbol{C}')_{\prec C} \rceil$, $V$ must in $\mathcal{G}_{\mathcal{S}}$ have an ancestor $A \in (\boldsymbol{X}' \cup \boldsymbol{C}')_{\succ C}$. Since $Z_0 \prec C$, $V$ this ancestor $A$ also has $Z_0 \prec A$. So, $A \in \mathrm{Desc}^{\mathcal{H}}(Z_0)$ by the definition of $\prec$, and $A \in \mathrm{Desc}^{\mathcal{G}}(Z_0)$ by the definition of $\mathcal{H}$, and $V \in \mathrm{Desc}^{\mathcal{G}}(Z_0)$, since $A$ is an ancestor of $V$.

Any fork in a path must either be an ancestor of the initial endpoint (in this case $Z$), or an ancestor of a collider in the path. Since $V \in \mathrm{Desc}^{\mathcal{G}}(Z_0)$ and $V$ is a fork, not an endpoint, $V$ cannot be an ancestor of the initial endpoint. So $V$ must be an ancestor of a collider in the walk $w$. As $w$ is active given $\lceil (\boldsymbol{X}' \cup \boldsymbol{C}')_{\prec C} \rceil$, the collider $D$ must be in $\lceil (\boldsymbol{X}' \cup \boldsymbol{C}')_{\prec C} \rceil$. We consider three sub-cases: 2.2.3.1: $D$ is in $\lceil (\boldsymbol{X}' \cup \boldsymbol{C}')_{\prec C} \rceil \setminus (\boldsymbol{X}' \cup \boldsymbol{C}')$, 2.2.3.2: $D$ is in $\boldsymbol{X}'_{\prec C}$, 2.2.3.3: $D$ is in $\boldsymbol{C}'_{\prec C}$, and will prove a contradiction in each case. *Sub-case 2.2.3.1: $D$ is in $\lceil (\boldsymbol{X}' \cup \boldsymbol{C}')_{\prec C} \rceil \setminus (\boldsymbol{X}' \cup \boldsymbol{C}')$.* Then all the parents of $\lceil (\boldsymbol{X}' \cup \boldsymbol{C}')_{\prec C} \rceil C$ must also be in $\lceil (\boldsymbol{X}' \cup \boldsymbol{C}')_{\prec C} \rceil C$ by the definition of implied variables, and these parents would be non-colliders, which would make $w$ blocked given $\lceil (\boldsymbol{X}' \cup \boldsymbol{C}')_{\prec C} \rceil C$, giving a contradiction. *Sub-case 2.2.3.2: $D$ is in $\boldsymbol{X}'_{\prec C}$.* Then at least one parent of $D$ must be a non-collider in $\boldsymbol{C}'_{\prec C}$, which contradicts the statement that $w$ contains no non-collider in $(\boldsymbol{X}' \cup \boldsymbol{C}')$. *Sub-case 2.2.3.3: $D$ is in $\boldsymbol{C}'_{\prec C}$.* Then $D \in \mathrm{Desc}^{\mathcal{G}}(Z_0)$ (since $D \in \mathrm{Desc}^{\mathcal{G}}(V)$ and $V \in \mathrm{Desc}^{\mathcal{G}}(Z_0)$). It follows that $Z_0 \prec D$, but this contradicts the definition of $C$ as the nearest variable along $w$ to $Z_0$ that is in $\boldsymbol{C}'_{\succ Z_0}$.

From Lemma 28 the result follows. $\qquad\square$