# OpenReview forum: "Toward a Complete Criterion for Value of Information in Insoluble Decision Problems"
_TMLR — Accepted by TMLR_

### Review · Reviewer_YwNK · 2024-07-31

**Summary Of Contributions:**

This paper investigates insoluble decision problems, where soluble graphs are those that retain important past observations. The authors theoretically establish the necessary conditions for the immateriality of insoluble graphs.

**Audience:**

No

**Broader Impact Concerns:**

I do not have any concerns.

**Claims And Evidence:**

No

**Requested Changes:**

Please address the weaknesses mentioned above, particularly in terms of improving the overall presentation of the paper and providing a clearer motivation for the problem being studied.

**Strengths And Weaknesses:**

### Strengths
- The necessary conditions for the immateriality of insoluble graphs have been theoretically analyzed with a detailed proof. This result can be valuable to the community focused on structural causal analysis and decision problems.
### Weaknesses
- The clarity of the paper is lacking, and the overall presentation quality is not high. Many sentences are difficult to understand due to various issues, including grammatical errors and insufficient explanations. Even the abstract is hard to comprehend, as it is written in an unusual manner with several grammatical mistakes. Since clarity of narrative and the strength of arguments are among the most important acceptance criteria, these aspects need significant improvement.
- The findings of this paper are quite specific. While I believe the paper holds some scientific value, considering another acceptance criterion—whether the findings will interest TMLR's audience—I feel that the scope is too narrow and specific. The content is difficult for a broad machine learning audience to understand, and the proof is highly technical. Therefore, a more specialized statistical journal might be a better fit for this paper. If the authors believe that their findings will appeal to a wider machine learning audience, this should be more clearly explained and emphasized in the paper.

---

> ### Comment · Action_Editor_oUzU · 2024-08-01
> **Thank you**
>
> Dear YwNK, thank you so much for your review

---

> ### Author Response · Authors · 2024-09-29
>
> Thank you for your comments, which we have used to improve the paper.
>
> # 1. Clarity.
> We agree that the paper had some grammatical errors and unclear wording. We have fixed the following (non-exhaustive):
>
> * P1: “The means that" $\to$ "This means that”
> * P2: “.Materiality?” $\to$ “. Materiality”
> * P3: “exogenous” $\to$ “endogenous”
> * P5: “satsify" $\to$ "satisfy”
> * P5: “$Y$ ,and” $\to$ “$Y$, and”
> * P5: “transmited” $\to$ “transmitted”
> * P6: “Three” $\to$ “Two”
> * P7: “hte policy" $\to$ "the policy”
> * P7: “in in Figure" $\to$ "in Figure”
> * P9: “no-longer" $\to$ "no longer”
> * P9: “incentivised X’ has to pass” $\to$ “incentivised X’ to pass”
> * P11: “assumptions of for” $\to$ “assumptions of”
> * P11: “seleciton” $\to$ “selection”
> * P13 “Each info path must pass on information from upstream paths that traverse the intersection node.” $\to$ “Each info path must transmit information from upstream paths that pass through the intersection node.”
> * P15: “the parent along the info path” $\to$ “its parent along the info path”
> * P16: missing comma inserted after if $P^\\pi(\\text{Pa}(Y^\\mathbf{p}_i) \\neq \\text{Pa}(T^{\\mathbf{p}_i}))<1$
> * P17: “satsify" $\to$ "satisfy”
> * P18: “When $\text{Pa}(Y^{\sp_i})=\text{pa}(Y^{\sp_i})$ and $u_{i,1}$ then $\text{Pa}(Y^{r_{i,1}})$ to assume one value” -> “When $\text{Pa}(Y^{\sp_i})=\text{pa}(Y^{\sp_i})$ and $U_{i,1}=u_{i,1}$, then $\text{Pa}(Y^{r_{i,1}})$ will assume one value.” ($\sp$ is bold math $p$ but there is a problem with markdown in OpenReview.)
> * P21: “non-observed variables would be needed to determine the value of $Z_i$” $\to$ “the value of $Z_i$ cannot be determined by observed variables”
> * P21: “Path-specific objectives”: noting AAAI publication.
>
> # 2. Interest to TMLR’s audience.
>
> We respectfully disagree that being narrow or specific precludes publication at TMLR. Rather, the acceptance criterion is whether some individuals in TMLR’s audience would be interested [https://jmlr.org/tmlr/acceptance-criteria.html]. We're pleased to note that reviewer eibw seems to disagree, noting that “the direction of this paper is undoubtedly important. This is a fundamental research question in causal models, and can have a great impact on various fields of machine learning”. We further emphasise that this direction was the main focus that featured in “Characterizing optimal mixed policies” (in NeurIPS 2020) and “A complete criterion for value of information in soluble influence diagrams” (AAAI 2022).
>
> We agree, however, that we can do more to highlight the potential of this work, and to do this, we have inserted an initial subsection to our literature review, as described to oetD.

---

### Review · Reviewer_eibw · 2024-08-22

**Summary Of Contributions:**

This paper considers whether a target value can be improved by interventions to structured causal models (Pearl, 2009). The set of possible interventions is specified by mixed policy scopes, which consist of variables to be intervened (decision variables) and variables that each decision variable can be based on (context variables). This intervention model follows the setting of Lee & Bareinboim (2020).

A goal of this research topic is to judge whether or not each context variable can improve the target value **only from the graph structure** of causal models. Precisely, given a context variable for some decision variable, the question is whether there exists a concrete probability distribution that aligns with the given causal graph in which an optimal decision based on the context variable is strictly better than an optimal decision that ignores the context variable. This property of the context variable is called "materiality" in this paper.

If there exists only one decision variable, materiality is known to be equivalent to a simple path-blocking property. On the other hand, if there exist multiple decision variables, a characterization of materiality is known only for a special class of graphs called "soluble graphs." This paper proves a sufficient (but not necessary) condition for materiality in general graphs. The proof significantly extends an idea of control paths, which connect context variables and the target variable without touching decision variables, and info paths, which connect decision variables and the target variables. For any graph satisfying the proposed sufficient condition, the authors construct a probability distribution in which observing each context variable improves the target value.

**Audience:**

Yes

**Broader Impact Concerns:**

The contribution of this paper is theoretical, and there is no ethical concern.

**Claims And Evidence:**

Yes

**Requested Changes:**

Minor requested changes:
- There are many small typos: "may used to" in the abstract, "hte" instead of "the", "in in Figure 4" , and many others. I recommend the authors to review the manuscript from beginning to end and fix them.
- The notation $\pi_X$ seems to be used in Definition 2 and Definition 7 for different meanings. It would be better to prevent this if it is possible.
- What is the meaning of "new parent" in the statement of Theorem 8?
- In 4.2.2, it is claimed that "A --> Z_0 has length of either 0 or 1" but actually not in Figure 6.
- In 4.3.2, what is the difference between $V_p$ and $V^p$?
- The sentence is incomplete with ending "so it instead" in page 18.

**Strengths And Weaknesses:**

Strengths:
- The paper is well-organized. In Section 3, the authors clearly explains the proof ideas of important previous studies that the proof of this paper is based on. Although the proof of the main theorem is extremely complicated, this organization helps readers understand the main proof ideas gradually. The concrete examples with readable figures are also very helpful.
- The direction of this paper is undoubtedly important. This paper aims at providing a necessary and sufficient condition for materiality. This is a fundamental research question in causal models and can have a great impact on various fields of machine learning (However, this paper provides only a partial result toward this goal).

Weaknesses:
- Although the organization is excellent, the writing quality of each sentence could be polished. As I write in Requested Changes, there are many small typos. Since the proof is complicated and requires readers to understand in detail, these small mistakes are confusing.
- This paper provides only a sufficient condition. I cannot judge how this result is close to a complete characterization. I am not sure if only a sufficient condition is worth the effort of such a complicated proof.

---

> ### Author Response · Authors · 2024-09-29
>
> Thank you for your comments, which have led us to make the following improvements to our paper:
>
> # Minor requested changes
>
> * Abstract: “sometimes may used to” $\rightarrow$ “may sometimes be used to”
> * Good point. We propose to replace the latter instance of $\pi$ with $\Pi$. This is a convention used in “Reasoning about Causality in Games” among other papers, and matches the idea that $V$ represents a variable, while $v$ represents its assignment.
> * Our initial mention of the “new parent” concept is in Def 7 on P8, and it is borrowed from Lemma 1 of Lee and Bareinboim (2020), where it appeared without explanation! But we agree that explanation is helpful, so we have added a footnote to P8: “\footnote{To be precise, each d-separation $\perp$ in (A-B) holds in the graph  $\mathcal{G}'$, obtained from $\mathcal{G}$ by adding a parent $\pi_X$ for each decision $X$.}”
> * P11 (Figure 6): we have extended the edge $A \to T_0$, so that the fact that this edge is dashed is now visible. Note that directed paths (represented by dashed lines) may in general have length zero, so this includes the case where $A=Z_0$.
> * P13 (4.3.2): $V_p$ changed to $V^p$.
> * P18: “Clearly this condition can be satisfied for any acyclic graph, so it instead” -> [merged with subsequent paragraph]

---

### Review · Reviewer_oetD · 2024-09-25

**Summary Of Contributions:**

This paper considers decision-making problems expressed by structural causal models.
In particular, it attempts to provide a characterization based on graph structure of variables that need to be observed for optimal decision making.
The characterization given in the previous study was limited to a special graph called soluble, whereas this study considers the case where this restriction is removed.
This paper shows that some of the conditions presented in existing studies are necessary conditions for this more general setting as well.
It also discusses the difficulty of identifying complete characterization through several specific examples.

**Audience:**

Yes

**Claims And Evidence:**

Yes

**Requested Changes:**

I would like to ask the authors for their views on the points listed under “Weakness” above.
In addition, the points listed in the “Minor comment” are not all that large, and I am concerned that there are many more points that need to be revised.

**Strengths And Weaknesses:**

Strength:
* The problem setting and its motivation are well explained.
* Various concrete examples and diagrams are used to illustrate the issues and contributions.

Weakness:
* Related studies and references cited in the paper seem limited.
This study is positioned as a follow-up to (Lee & Bareinboim, 2020).
The only other comparisons mentioned are mainly (Everitt et al., 2021), (Farquhar et al., 2022), and (Merwijk et al., 2022).
If these are all the major relevant studies, the number of readers interested in this paper may be limited.
It was also surprising that while the problem setup seemed natural, previous studies were limited to these.
If the authors have any thoughts on why there is limited research on the problem setting (or similar ones) in this paper,
I would like to see their explanation.
* Many roughnesses and typos are found in the textual expression.
For example, p. 20 concludes with the following sentence, which I think is difficult to understand what exactly is being asserted.
> It seems that new insights are needed to solve this superimposition problem, and that therefore that we will need new insights to establish a complete criterion for materiality in insoluble decision problems.

Other issues of presentation are listed in [Minor comments].
* It is difficult to interpret the claim of the main result (Theorem 8) well. In particular, since C in Theorem 8 is apparently different from that under the previous study, it would be better to explain how this can be interpreted.


[Minor comment]
* p.1, Abstract: may used to evaluate ... <- may be used to evaluate ... ?
* p.1, Abstract: as immaterial For ... <- as immaterial. For ...
* p.1, last sentence in the second paragraph in Introduction: The means <- This means
* p.1, last sentence in the second paragraph in Introduction: $1 - 0.5 = 0.,$ <- $1 - 0.5 = 0.5,$
* p.2, first sentence in the last paragraph: there the criterion <- there is the criterion
* p.3, first sentence in the fourth paragraph: (2020, thm. 2) <- (2020, Thm. 2)
* p.4, Definition 2: Symbols of the domain of $\pi$ are not defined?
* p.4, sentence just before Definition 3: Once a policy is selected <- Once a mixed policy scope is selected ?
* p.5, last sentence: $Y$ ,and <- $Y$, and
* p.7, second sentence of the second last sentence: in in Figure 4 Suppose <- in Figure 4. Suppose
* p.8, second sentence: in this paper <- In this paper
* p.18, second last paragraph: so it instead ...?
* p.19, third paragraph in the last paragraph of Section 5.1: two definitions of $C'$

---

> ### Author Response · Authors · 2024-09-29
>
> We thank the reviewer for highlighting the quality of our explanation of the problem setting, its motivation, and the clarity of our examples. We now address the three areas where the paper might be improved.
>
> ## 1. Relevant studies and number of interested readers
>
> We agree that it is worth doing more to highlight the range of relevant studies beyond those of Lee, Bareinboim, van Merwijk and Everitt. We have therefore inserted a new subsection of Section 3 called “Graphical criteria for materiality, and their applications”. This subsection highlights that: 1) value of information has been of interest for a long time, 2) a number of studies have tried to develop graphical criteria for materiality, and 3) there are some key application areas, such as evaluating agent’s incentives, including for the purpose of developing fair and safe AI systems.
>
> ## 2. Clarifying the difference between Theorem 8 and the previous study.
>
> To begin with, we note that the relationship to previous work is described above Theorem 8: we prove that condition (I) of LB-factorisability is necessary to establish immateriality, while leaving open the necessity of conditions (II-III). We agree that further explanation is needed, so we have added the following sentence *below* Theorem 8:
> > “In words, this means that if each variable provides information about the outcome given other contexts (condition B), as well as all decisions, and everything determined by them (condition C), and moreover that this outcome is influenceable (condition A), then each variable will be material in at least one model compatible with the graph.”
>
> ## 3. Minor comments
>
> Thank you very much for these typo corrections - we have implemented all of them.

---

> ### Author Response · Authors · 2024-09-29
>
> We have also improved the precision of the last sentence of Section 5 (p20), revising it as follows:
> "Overall, in order to establish a complete criterion for materiality, we would need some new method to prevent
> the information from Z_0 from being superimposed on other decisions. So, in order for future work to achieve
> this goal, we predict that it will have to make further modifications to the construction from Definition 16."

---

### Decision · Action_Editor_oUzU · 2024-11-05

**Recommendation:** Accept with minor revision

**Comment:**

The tackled problem is relevant and the proposed approach is sound. However, as pointed out by some of the reviewers I think there are concerete concerns about the clarity of the writing which has been partially addressed in the revision. I suggest the authors to go through the comments from the reviewers and to address the main concern about the clarity of the writing of the manuscript.

**Audience:**

I agree with the opinion of all reviewers that the the topic and its development is of interest to the TMLR audience.

**Claims And Evidence:**

According to all three reviewers the manuscript supports all the claims made by convincing and clear evidence.

---

> ### Author Response · Authors · 2024-12-03
> **camera-ready version**
>
> Dear editor,
> Thanks for your recommendation for acceptance with minor revision.
>
> We believe that the specific concerns about clarity have been addressed in our previous revision, according to the line-by-line responses to the various reviewers.
>
> In addition, to address the general concern about clarity, we have reread and edited the paper in its entirety, including many typo corrections, rewordings, and reorderings of arguments, in order to improve readability, while keeping the overall organisation of the paper unchanged.
> We also added clarifying explanations to Definition 15 and Lemma 23 in particular.
>
> If any further changes are required, please let us know
> Best,
> Authors